# Assessing the impact of SARS-CoV-2 prevention measures in Austrian schools using agent-based simulations and cluster tracing data

Jana Lasser [1,2✉], Johannes Sorger[2], Lukas Richter [3,4], Stefan Thurner [2,5,6], Daniela Schmid[4] & Peter Klimek [2,5✉]

We aim to identify those measures that effectively control the spread of SARS-CoV-2 in Austrian schools. Using cluster tracing data we calibrate an agent-based epidemiological model and consider situations where the B1.617.2 (delta) virus strain is dominant and parts of the population are vaccinated to quantify the impact of non-pharmaceutical interventions (NPIs) such as room ventilation, reduction of class size, wearing of masks during lessons, vaccinations, and school entry testing by SARS-CoV2-antigen tests. In the data we find that 40% of all clusters involved no more than two cases, and 3% of the clusters only had more than 20 cases. The model shows that combinations of NPIs together with vaccinations are necessary to allow for a controlled opening of schools under sustained community transmission of the SARS-CoV-2 delta variant. For plausible vaccination rates, primary (secondary) schools require a combination of at least two (three) of the above NPIs.

[1] Institute for Interactive Systems and Data Science, Faculty of Computer Science and Biomedical Engineering, Graz University of Technology, Rechbauerstrasse 12, 8010 Graz, Austria. [2] Complexity Science Hub Vienna, Josefstädterstrasse 39, 1080 Vienna, Austria. [3] Institute for Statistics, Faculty of Mathematics, Physics and Geodesy, Graz University of Technology, Rechbauerstrasse 12, 8010 Graz, Austria. [4] Österreichische Agentur für Gesundheit und Ernährungssicherheit GmbH, Spargelfeldstrasse 191, 1220 Vienna, Austria. [5] Section for Science of Complex Systems, Center for Medical Statistics, Informatics and Intelligent Systems, Medical University Vienna, Spitalgasse 23, 1090 Vienna, Austria. [6] Santa Fe Institute, 1399 Hyde Park Road, Santa Fe, NM 87501, USA. ✉email: jana.lasser@tugraz.at; peter.klimek@meduniwien.ac.at

As SARS-CoV-2 transitions into endemicity, returning teachers and students to schools safely requires a precise public health approach that is guided by evidence on the actual transmission dynamics in different types of schools, and a thorough evaluation of the effectiveness of different non-pharmaceutical interventions (NPIs) to prevent in-school-transmissions[1–4]. Currently available evidence suggests a relatively low transmission of SARS-CoV-2 in schools[5–11], particularly among younger students[12–15]. However, outbreaks with attack rates of up to 17%[16] have been reported particularly in secondary school settings and in regions with a high SARS-CoV-2 incidence in the community[17,18]. In addition, the now dominant B1.617.2 virus strain (delta variant) poses additional challenges due to its drastically increased transmissibility[19,20] and shorter incubation period[21], especially in situations where younger children are not eligible for vaccination and virus circulation in the population is high.

Most countries are currently deploying a range of different NPIs to prevent transmissions in schools. In addition to an increasing prevalence of vaccinations in the population, potential containment measures include wearing of masks (also inside class rooms), class size reductions through student cohorting, and room ventilation. With the emergence of new generations of antigen (AG) tests, at-home or self-testing is now also becoming feasible at scale and allows for screening strategies in schools to rapidly identify asymptomatic or presymptomatic cases of infection[22–24]. Modelling studies suggest that the low sensitivity of AG tests, compared to PCR tests, can be offset by their short turnover time (the test result is available within minutes) and more frequent use[25].

To date there is limited evidence of how effective these measures or their combinations are to prevent transmissions in different school types in a situation where the delta variant is dominant and only parts of the population are vaccinated. Descriptions of clusters in schools are often limited to a handful of outbreaks[26–28] or do not reliably delineate in-school transmissions from out-school transmissions among school-aged children[29]. While sophisticated modelling approaches for the transmission dynamics in schools have been proposed[30–32], these models often lack the comprehensive and detailed information on a large number of school clusters that would be needed for a proper calibration and validation. In addition, the emergence of new variants of concern with different characteristics emphasizes the importance to adapt models to current circumstances.

Here, we develop an agent-based model that is calibrated to Austrian data on school-clusters, in order to evaluate the effectiveness of combinations of NPIs in preventing transmission in six different school types: primary, lower secondary, upper secondary, secondary, with or without day care. A cluster is defined as a group of at least two cases of SARS-CoV-2 infection, which were epidemiologically linked as an infector (i.e. source case) and an infectee (i.e. successive case). A school cluster includes at least one infectee generated by in-school transmission. The source case of the in-school transmission(s), a teacher-source case or a student-source case, occurs in any other setting, such as household, work place, leisure activity, or in an unknown setting. For identifying the source case and successive cases, we used information on disease onset and possibly contagious interactions within 14 days prior to disease onset. These data are derived from standardised case-interviews performed by the responsible public health authorities. For the model we included 616 clusters involving 2,822 student-cases and 676 teacher-cases that occurred between calendar weeks 36 and 45, in 2020. The model couples in-host viral dynamics with population dynamics taking place on contact networks determined by school type, number of classes, and average class size. The properties of these contact networks are modelled from timetables and educational statistics for

Austrian schools of different types; they are time-dependent (students and teachers follow a weekday-specific schedule where contacts take place in classes, teacher facilities, during daycare, or between siblings in households) and multi-relational (transmission risks depend on intensity and type of contact). The viral dynamics allow for a more faithful representation of testing strategies. The probability of an exposed individual to transmit the disease, as well as the probability to be tested positive changes over the course of an infection and, additionally, depends on the presence of symptoms.

The model is calibrated to actual Austrian cluster data to ensure that it reproduces (i) realistic cluster sizes and (ii) the ratio of infected teachers to infected students. Model inputs include the empirically observed age-specific ratio of symptomatic to asymptomatic infected individuals and the ratio of teacher-to-student source cases as a function of student age. We calibrate the transmission risk for contacts in households, schools, and their age dependence to match characteristics of outbreaks from autumn 2020 when the original strain of the virus was dominant and no vaccine was available.

To adapt the calibrated model to a situation in which the delta variant is prevalent, we multiply the household transmission risk—which was calibrated using data from outbreaks of the original strain—by 2.25, reflecting the drastically increased infectivity of the variant[19,20] and adapt the incubation period to $4.4 \pm 1.9$ days[21]. We also adapt the latent period to $3.0 \pm 1.9$ days.

Using the model, we study the effectiveness of four categories of NPIs and their combinations as well as different shares of vaccinated students or teachers. The NPIs include (i) room ventilation, (ii) use of surgical masks of teachers and students during lessons (it is assumed that masks are always worn in schools when students are not sitting at their desk), (iii) student cohorting, and (iv) screenings by use of entry AG tests for SARS-CoV-2 at different frequencies. As a reference scenario we consider a test-trace-isolate strategy in which students and teachers are PCR-tested as soon as they develop symptoms. We evaluate to which extent each of these measures contributes separately to a reduction of cluster size with respect to the reference scenario. We then assess the combined effectiveness of NPI bundles and the sensitivity of our results with respect to the stringency of implementing these measures and for different vaccination rates. Our aim is to quantify how many transmissions can be expected for the different scenarios in the different school types, in a way that is appropriate to derive evidence-based policies for keeping schools open at a controllable infection transmission risk. We note that out-of-school mixing is beyond the scope of our analysis.

## Results
**Cluster analysis**. We identified 616 clusters including at least one in-school transmission in the data. In total, these clusters involved 9232 cases. Out of these, 2822 were student-cases and 676 teacher-cases, of which 464 were source cases (introduced the virus into the school setting) and 3034 cases were generated by in-school transmission. In total, 286 cases were related to primary schools (69% students), 762 to lower secondary schools (79% students), 388 to the upper secondary schools (89% students) and 810 to the secondary schools (88% students), see also Fig. 1B. The portion of student-source cases was lowest in primary schools (6%), followed by lower secondary (43%), secondary (64%) and upper secondary (82%), see Fig. 1A. The average size (i.e. number of cluster cases) of clusters with a teacher as source case (5.7 cases) was larger than that of clusters with students as source (4.4 cases). The clinical presentation was clearly age-dependent. While

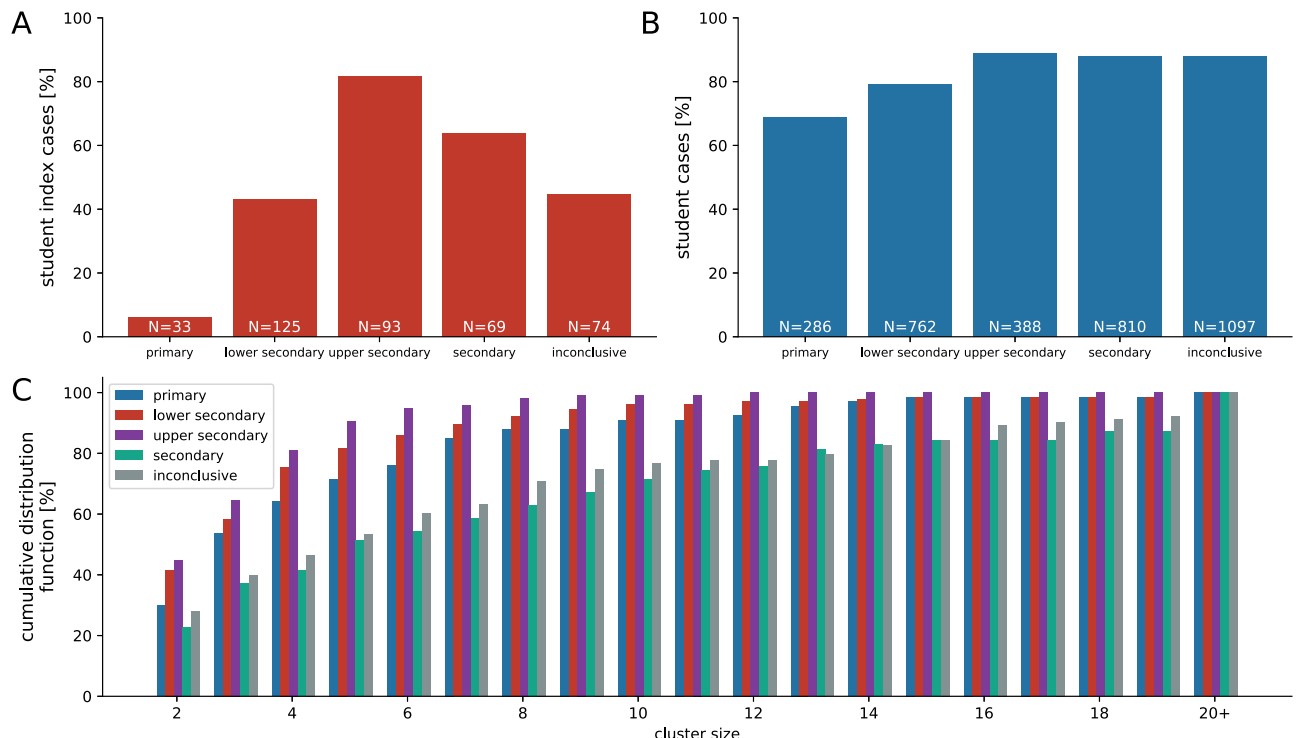

**Fig. 1 Cluster data for transmissions in schools.** The ratio of (**A**) student-to-teacher source cases is lower in school types with young children (primary, lower secondary) and higher in school types with older children (upper secondary, secondary). The same trend is seen in the proportion of students in clusters (**B**). **C** The cumulative distribution function of cluster sizes shows a majority of clusters consisting of only a handful of cases, whereas clusters with 20 or more cases still do occur, particularly in (upper) secondary schools. See Methods Section "Empirical observations of SARS-CoV-2 clusters in Austrian schools" for details on the identification of school types.

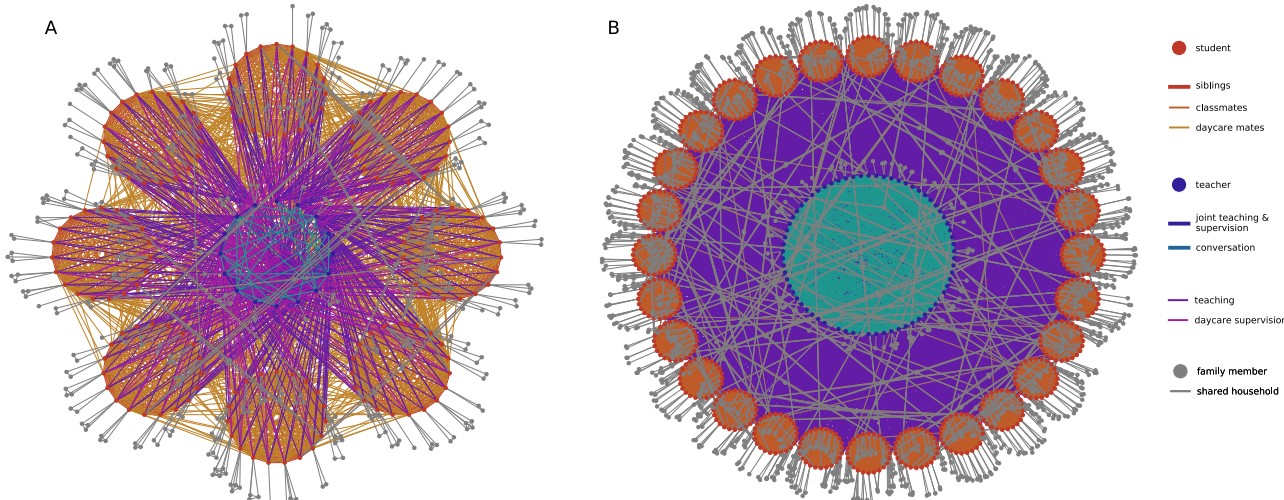

**Fig. 2 Contact networks of agents in schools. A** Representative Austrian primary school with afternoon daycare, including 8 classes with 19 students each, 16 teachers, and their respective family members. **B** Typical Austrian secondary school, including 28 classes with 24 students each, 70 teachers, and their respective family members. Contacts between students (red circles) are depicted in shades of red and include contacts between class mates, students in the same daycare supervision group and siblings. Contacts between teachers (blue circles) are depicted in shades of blue and include contacts between teachers that teach or supervise together or have meetings. Contacts between household members (grey circles) are depicted as grey lines. Contacts between teachers and students that arise during teaching or daycare supervision are depicted as purple and pink lines.

4 out of 6 students younger than 6 years old were asymptomatic, the proportion of asymptomatic cases dropped by increasing age from 61%, 49%, 33% and 16% for the age groups 6–10, 11–14, 15–18 and adults (including students and teachers), respectively. Figure 1C shows the distribution of cluster size by different school types. Overall, cluster sizes of 2, 3–9, 10–19 and 20+ cases accounted for 40%, 49%, 8% and 3%, respectively.

**Model calibration**. We depict two exemplary contact networks for the typical Austrian primary and secondary school types in Fig. 2 that were modelled using data from the Austrian school statistics[33], time tables and stakeholder interviews (see Methods Section "Interviews with school personnel" for details). Contact networks vary substantially between primary and secondary schools, the latter showing larger and denser networks. Increased

**Table 1 Mechanisms $q_i$ that modify the probability of a successful transmission, $p$, between agents.**

| Variable | Mechanism | Values | Source |
|---|---|---|---|
| $q_1$ | Contact type | Household: 0, school: 0.7 | Calibration |
| $q_2$ | Age of transmitting agent | see Eq. (1) | Calibration |
| $q_3$ | Age of contracting agent | see Eq. (2) | Calibration |
| $q_4$ | Infection progression | see Eq. (3) | 45,49 |
| $q_5$ | Symptoms | Symptomatic: 0, asymptomatic: 0.4 | 50 |
| $q_6$ | Surgical mask transmitting agent | No mask: 0, mask: 0.5 | 51 |
| $q_7$ | Surgical mask contracting agent | No mask: 0, mask: 0.3 | 51 |
| $q_8$ | Room ventilation | No ventilation: 0, once per hour: 0.64 | 37 |

Values for the respective modifications are either calibrated using data of clusters that occurred in Austrian schools or taken from literature.

network density is driven by teachers playing a more relevant role as links between classes in secondary schools due to differences in the structure of curricula. In secondary schools, teachers often change between class rooms during the day whereas in primary schools they typically supervise only one class. To reproduce the empirically observed cluster characteristics, we find the transmission risk for school contacts $c_{contact}$ to be 70% [66%; 74%] (2.5 and 97.5 percentile values) lower than the transmission risk of household contacts which per calibration is found to be 7.37 ± 0.02% (mean ± SD) per day to transmit an infection. See also Table 3 for contact classifications. Children are 0.5% [0%; 2.25%] less likely to transmit an infection per year they are younger than 18 years. Their transmission risk is reduced by $-c_{age}(18-y)$%, where $y$ is age and $c_{age} = -0.005$. The 2.5 and 97.5 percentile values of $c_{age}$ are 0.0 and $-0.0225$, respectively. If not stated otherwise, we use $c_{contact} = 0.3$ and $c_{age} = -0.005$ for all subsequent simulations to analyze the effect of different prevention strategies on characteristics of clusters in schools. Sensitivity of the model with regards to our calibration parameters is low: If the transmission risk for school contacts is [66%; 74%] lower than that of household contacts, average outbreak sizes over all school types and a range of scenarios (see Supplementary Fig. 3 for details) change by a factor of [0.96; 1.05]. Changes in outbreak sizes for primary and secondary schools are not significantly different with a factor of [0.97; 1.04] and [0.96; 1.05], respectively.

If the age dependence of the transmission risk $c_{age}$ is [−0.0225; 0.0], average outbreak sizes change by a factor of [0.92, 1.03], where the change in outbreak sizes in primary schools is slightly more pronounced ([0.89, 1.04]) than the change in secondary schools ([0.96, 1.02]). These changes are small when compared to changes in assumed measure effectiveness (see section Sensitivity Analysis below).

Parameter values for the effectiveness of individual measures are taken from the literature; see Table 1. To study sensitivity of our results to these values, we also investigate how they change for more conservative estimates.

**Effectiveness of measures.** Results for the effectiveness of individual measures in the absence of vaccinations are shown in Fig. 3 in terms of the distributions of cluster sizes with students or teachers as source cases. Next to the distributions we show mean values of the reproduction number $R$ measured for the source cases in the simulation. In Supplementary Table 1, we additionally show the average cluster size along with the 75th and 90th percentile of clusters with students and teachers as source cases, respectively. If $R < 1$, we consider a cluster in a given scenario as "controlled". See Methods Section "Calculating the reproduction number, R" for the calculation and interpretation of $R$.

As reference we consider a "no mitigation" scenario in which no preventive measures except diagnostic testing and screening in the event of a symptomatic case (see Methods Section "Testing

and Tracing") are enacted. This assumption results in a bimodal distribution of the cluster size containing small clusters of ten or less cases and large cluster with up to hundred cases or more in secondary schools. It is important to note that a significant number of source cases do not spread the infection further and do not result in a cluster. Depending on the school type, this can be true for up to 63% of source cases (primary schools).

Assuming average school sizes of 152, 144, 230 and 674 students for primary, lower secondary, upper secondary and secondary schools[33] and no vaccinations, on average we find 69 cases per cluster in primary schools (75th percentile: 127 cases, 90th percentile: 206), 211 (303, 3014) cases in lower secondary schools and 1073 (1402, 1420) cases per cluster in secondary schools, overall. For each school type the average reproduction number is larger than one, ranging from 2.6 (standard deviation 2.1) in primary schools, and 3.2 (SD 2.4) in lower secondary schools to 3.6 (SD 2.7) in secondary schools if the source is a student. Clusters with teachers as source case are typically larger and show higher reproduction numbers, ranging from 4.4 (SD 2.9) in primary, 8.1 (SD 4.5) in lower secondary to 9.8 (SD 6.1) in secondary schools.

In the following, we focus on results obtained for primary and secondary schools. Primary and secondary schools typically show the smallest and largest outbreak sizes, respectively. Upper secondary schools tend to have a slightly higher reproduction number than secondary schools (but are smaller in size). Next to their numbers of classes, students and teachers, see Table 2, the transmission dynamics differs across school types due to different network densities (students and teachers have average degrees of 23.6 ± 0.1 and 43.1 ± 0.4, respectively, on school days in primary schools, and 33.6 ± 0.1 and 99.6 ± 0.2 in secondary schools), and the age-dependent transmission risk. While the latter accounts for changes in transmissibility of maximally 6%, average network degrees vary by up to 230%.

Considering each measure separately, we see the largest reduction in cluster size for room ventilation (estimated 64% reduction in transmission risk; see Methods Section "Transmission risk"). Screening (see Methods Section "Testing and tracing" for how test sensitivity as a function of viral load was modelled) by means of active case finding through AG testing is the most effective NPI. Testing students twice a week reduces the cluster size to 3 (3, 6) [14 (18, 33)] cases in primary and 128 (5, 695) [600 (726, 752)] in secondary schools when students [teachers] are the source case.

Testing teachers twice a week reduces cluster size, for student [teacher]-source clusters to 27 (39, 79) [13 (6, 45)] in primary and 762 (1174, 1218) [503 (1142, 1204)] in secondary schools.

Room ventilation has the second biggest impact, with 6 (7, 14) [9 (11, 22)] cases in primary schools, to 240 (653, 749) [461 (729, 784)] cases in secondary schools.

If the size of classes is reduced (alternately removing all school-related contacts of 50% of students on every second school

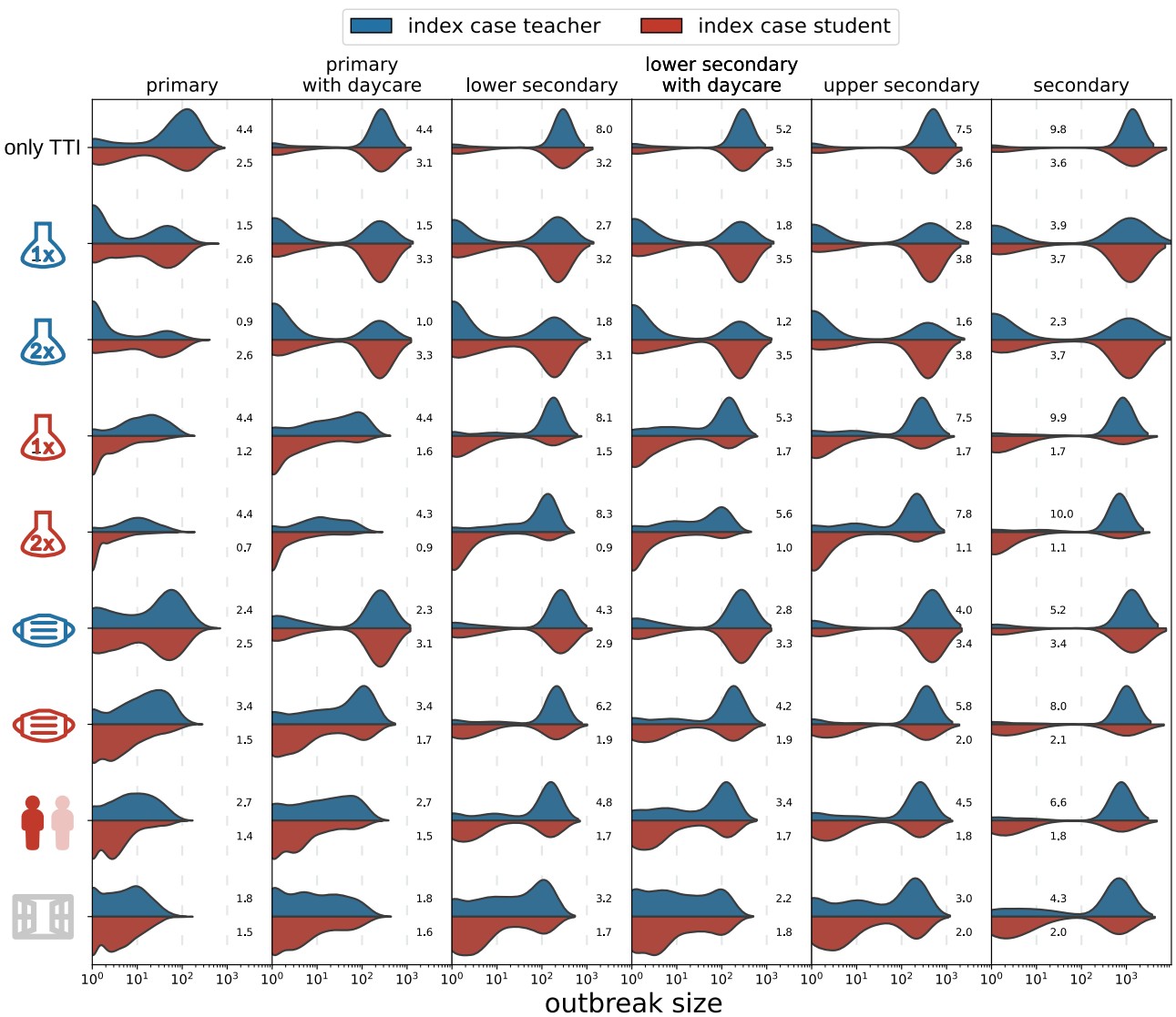

**Fig. 3 Distributions of cluster sizes for different school types (columns) and measures (rows) with teachers (blue) or students (red) as the source cases.** Results are shown for only test-trace-isolate (TTI) as baseline and TTI plus an individual measure (testing teachers/students once or twice a week, teachers or students wearing masks during the lessons, class size reductions, ventilation), respectively. Mean values of $R$ are shown next to the distributions.

**Table 2 Number of classes, students and teachers of typical schools of different school types in Austria.**

| School type | # classes | # students | # teachers |
|---|---|---|---|
| Primary | 8 | 19 | 12 |
| Primary with daycare | 8 | 19 | 16 |
| Lower secondary | 8 | 18 | 20 |
| Lower secondary with daycare | 8 | 18 | 24 |
| Upper secondary | 10 | 23 | 29 |
| Secondary | 28 | 24 | 70 |

The number of teachers was determined based on the allocation formula per school type as well as interviews with teachers and principals.

day; see Methods Section "Contact networks") cluster sizes are 6 (5, 13) [14 (19, 34)] in primary schools and 278 (755, 810) [666 (802, 836)] cases in secondary schools for students [teachers] as source cases. If students wear masks (estimated to reduce transmission risks by 50% and 30% if the transmitting and contracting agents wear a mask, respectively; see Methods Section

"Transmission risk") also during lessons when sitting at their desk, the cluster sizes are reduced to 9 (8, 25) [23 (36, 56)] cases in primary and 438 (1006, 1053) [905 (1041, 1070)] in secondary schools with students [teachers] as source cases.

To summarize, of all single NPIs, only 2-times testing among students is sufficient to reduce the average reproduction number for student-source clusters below one in primary and lower secondary schools. In all other school types and all other considered scenarios, control of SARS-CoV-2 spread in schools in the sense of of $R < 1$ requires a combination of more than one preventive measure.

**Combination of measures**. Results for selected combinations of preventive measures in the absence of vaccinations are shown in Fig. 4 and Table 2.

We first consider combinations in which we sequentially combine NPIs in the following order: room ventilation, mask usage by teachers, mask usage by students, and class size reduction. This roughly corresponds to an increasing complexity and cost of measure implementation in practice. In a scenario

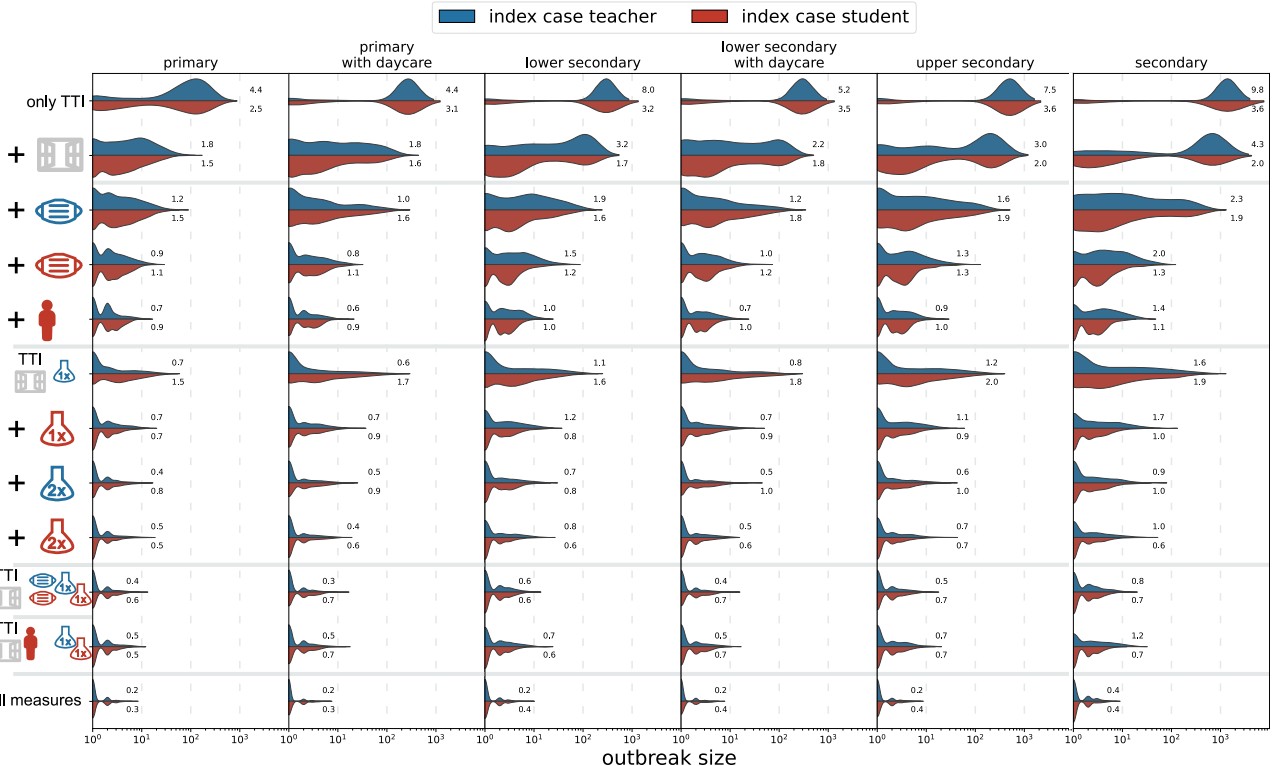

**Fig. 4 Distribution of cluster sizes as in Fig. 3 for combinations of measures.** From top to bottom, we first consider only test-trace-isolate (TTI) (first row) plus ventilation (second row). We then evaluate the combination of TTI and ventilation with masks for teachers, masks for students, as well as class size reductions. Next, we evaluate the combination of TTI and ventilation with test screenings amongst teachers and students once or twice per week. Cluster sizes are even further reduced for measure combinations that combine ventilation and testing with masks or class size reductions. Finally, we evaluate the combination of all of the above measures. Mean values of $R$ are shown next to the distributions.

where all NPIs except for preventive testing are implemented, $R$ drops below one only in primary schools, with outbreak sizes of 2 (3, 4) and 2 (2, 4) for student- and teacher-source cases, respectively. For all other school types, in general some form of preventive testing is necessary to achieve $R < 1$.

For combinations of measures including room ventilation together with different preventive test strategies in primary schools, ventilation and testing teachers and students only once a week by AG tests achieves $R < 1$ for teachers and students as source case.

Controlling clusters which originate from students can be achieved through weekly testing of teachers and students for SARS-CoV-2 infection for primary and lower secondary schools. Upper secondary and secondary schools require at least two preventive tests of all students per week.

Room ventilation and weekly AG testing combined with mask-usage result in spread control of clusters with a teacher-source case in all school types.

Combining ventilation and testing with class size reduction shows a smaller outbreak size reduction compared to mask-usage, e.g., with an average cluster size of 3 (4, 8) and $R = 1.2$ (SD 1.5) for clusters with a teacher-source case in secondary schools. Combining all aforementioned NPIs results in $R < 1$ for all school types and both types of cluster source-case.

**Sensitivity analysis.** The effectiveness of every preventive measure depends on the implementation in practice. Therefore, we systematically vary the parameters determining NPI effectiveness in the simulations and compare the resulting cluster sizes to the cluster sizes of the "baseline" scenarios shown in Fig. 4. Regarding the efficiency of screening for SARS-CoV-2 infection among

asymptomatic students and teachers by AG testing, we vary the sensitivity between 10% and 100%, the proportion of teachers and students participating voluntarily in testing between 10% and 100% of the agent population and the proportion of students staying at home when class sizes are reduced between 20% and 80% in each class. We vary the transmission risk reduction that can be achieved by room ventilation between 20% and 90% and the reduction associated with mask-usage between 30%, [10%] and 90% [70%] for exhaling [inhaling]. In addition, we relax the assumption that there are no contacts between students of different classes through friendships or meetings during breaks or lunch. To account for this, we add additional contacts between a random number of students from any given class to random students from other classes. We vary the ratio of students that have such contacts between 0% and 40%.

To illustrate the combination of the different efficiencies of the measures of interest, we assume a "worst case" scenario with conservative estimates for the parameters. Thus, we consider a sensitivity of 40% for the AG test (as compared to 100% in the optimal scenario, see also Methods Section "Testing and Tracing"), a participation of teachers and students in AG testing of 50% (as compared to 100% in the optimal scenario). This small proportion of participating persons may reflect the phenomenon of increased "COVID skepticism" and the concerns voiced by parents about their children being tested at school[34].

Furthermore, we consider for class size reduction that 30% of students stay at home (compared to the targeted 50%). In order to reflect the uncertainty of room ventilation on reduction of aerosols, we consider a transmission risk reduction of 20% (as compared to 64% in the optimal scenario), see for example Curtius et al. 2021[35], Sun and Zhai[36], Lelieveld et al. 2020[37]. We consider mask-wearing during lessons to be associated with a

transmission risk reduction of 40% and 20% for exhaling and inhaling, respectively (as compared to 50% and 30%, respectively, in case of optimal mask efficiency). Finally, we introduce additional between-class contacts for 20% of the students. In summary, these estimates represent informed guesses, as to date there are no reliable data available to assess these parameters.

In addition to the average number of transmissions of the source case, $R$, we report results as the "fold-increase", $X$, i.e., the factor by which the average cluster size of a given scenario is increased with respect to the average cluster size of the baseline scenario (given in Supplementary Table 2). Results are shown in Fig. 5A for the same measure combinations that we assessed in Fig. 4. We assess the varying efficiency of a single measure in Supplementary Figs. 4 and 5.

In all scenarios, the reduced effectiveness of the preventive measures leads to an exponential increase of the cluster size, demonstrating substantial sensitivity with respect to the estimated measure efficiencies.

For preventive testing strategies in the conservative scenario (without any vaccinations, see Fig. 5, column A), we observe a more than 270-fold increase in outbreak sizes in secondary schools in a scenario where teachers and students are tested once per week, and a 17-fold increase in primary schools. Combining testing (2x) with class size reduction shows an increase of the average cluster size of a factor of four in primary schools and still 157-times in secondary schools for clusters with students as the source case. Preventive strategies that combine mask-wearing with either reduced class sizes or testing (2x) show smaller increases in cluster sizes ranging from 1.7- to 3.5-fold in primary schools and 27- to 59-fold in secondary schools. In summary, with these very conservative assumptions for measure effectiveness, no combination of measures is able to achieve $R < 1$, even in primary schools. This means that control of the delta variant in schools cannot be achieved by these NPIs under conservative assumptions in an unvaccinated scenario.

**Vaccinations**. We consider two vaccination scenarios: In vaccination scenario I, 80% of teachers, 60% of family members reflecting the vaccination prevalence in the general population and 0% of students are vaccinated (in line with observations of vaccine uptake in Austrian teachers) (see https://orf.at/stories/3227808/, accessed on September 12, 2021) and the general population (see https://info.gesundheitsministerium.at/, accessed on September 12, 2021) as of September 2021. In vaccination scenario II, an additional 50% of students are vaccinated.

In Fig. 5 (columns B and C), we report the impact of different combinations of measures on outbreak sizes assuming conservative estimates for measure stringency under both vaccination scenarios. For the first scenario (no students vaccinated), we find that for student source cases a combination of ventilation, masks and the reduction of class sizes is sufficient to achieve $R < 1$ for all primary and lower secondary schools and $R > 1$ for upper secondary and secondary schools for student source cases. For teacher source cases, we have $R > 1$ for all tested measure combinations. Outbreak sizes range from 2 (3, 5) [4 (5, 8)] for student [teacher] source cases in primary schools, and 4 (4, 8) [12 (14, 28)] in secondary schools.

If an additional 50% of students are vaccinated and all available NPIs are implemented, we find $R < 1$ for all school types and source cases except for teacher source cases in secondary schools ($R = 1.3(1.4)$). In this scenario, outbreak sizes range from 2 (2, 3) [2 (2, 4)] in primary schools to 2 (2, 4) [4 (5, 8)] in secondary schools. Results for all measure combinations and both vaccination scenarios are given in Supplementary Tables 4 and 5.

The effect of vaccinations can also be observed in the ratio of infections in schools relative to households. For instance, in the no mitigation scenario we find on average 1.10 infections in household members for each identified case in primary schools and 1.01 infections for secondary schools. In vaccination scenario I this reduces to 0.39 and 0.35 infections, respectively; in the second vaccination scenario we find a similar number of household infections per school case.

**Online visualization**. To communicate our results to stakeholders (i.e., school administrative staff, parents, as well as students) and allow them to investigate the effect of measures within their specific school setting, we created an online simulation viewer (see https://vis.csh.ac.at/covid-schools/) as an interactive interface to our study results. The tool allows users to configure a school in terms of the school type (see Supplementary Information Section 1), class size (i.e., average number of students), number of classrooms, and number of floors. Based on their configuration, users receive an overview of the effect of measures on their schools in terms of cluster size and resulting quarantine days (see Methods Fig. 9). Individual measure configurations can then be selected to display an animation that illustrates a representative cluster development in the school over time (see Methods Fig. 10). We note that simulations in the online simulation viewer are intended for educational purposes only and still reflect infection dynamics based on the original strain and do not incorporate changes introduced with the delta variant as of September 2021.

**Discussion**
We analyzed Austrian data on 616 SARS-CoV-2 clusters with at least one in-school transmission. We used this data to calibrate an agent-based epidemiological model that quantifies the effectiveness of combinations of preventive measures across different school types for student-source cases and teacher-source cases. Different types of schools require different preventive measures to control the spread of SARS-CoV-2. As any single measure (mask-wearing during lessons, room ventilation, school entry testing, class size reduction) is typically not enough to achieve control ($R < 1$), the school management needs to think in terms of smart combinations of measures to safely operate schools in the COVID-19 pandemic. As parameter choices for the effectiveness of individual prevention measures are based on sometimes preliminary literature, we note that our findings need to be interpreted with care, in particular if new related evidence should arrive.

Current evidence suggests that schools mirror the infection dynamics observed in the general population[17,18], though it seems that smaller children typically contribute less to virus spread[12–15]. In line with these findings, we find that the setting-specific reproduction number for clusters with teachers as source case increases from around 4.4 for primary schools to more than 9.8 for secondary schools in a scenario with TTI only, whereas for students as source case the reproduction numbers range between 2.6 and 3.6. Secondary schools are therefore a riskier transmission setting, particularly if a contagious teacher is present. Keeping schools open in a controlled way in regions with sustained community transmission is therefore only feasible if stringent mitigation measures are put in place and are strictly adhered to[38].

We find two main reasons for the differences between primary and secondary schools. Even if we assume viral load in children to be comparable to adults, less coughing, smaller lung volume, and the emission of aerosols from a lower height with suspension in the air for a shorter duration of time suggest that transmission risk increases with age in children. Indeed, our calibrated model

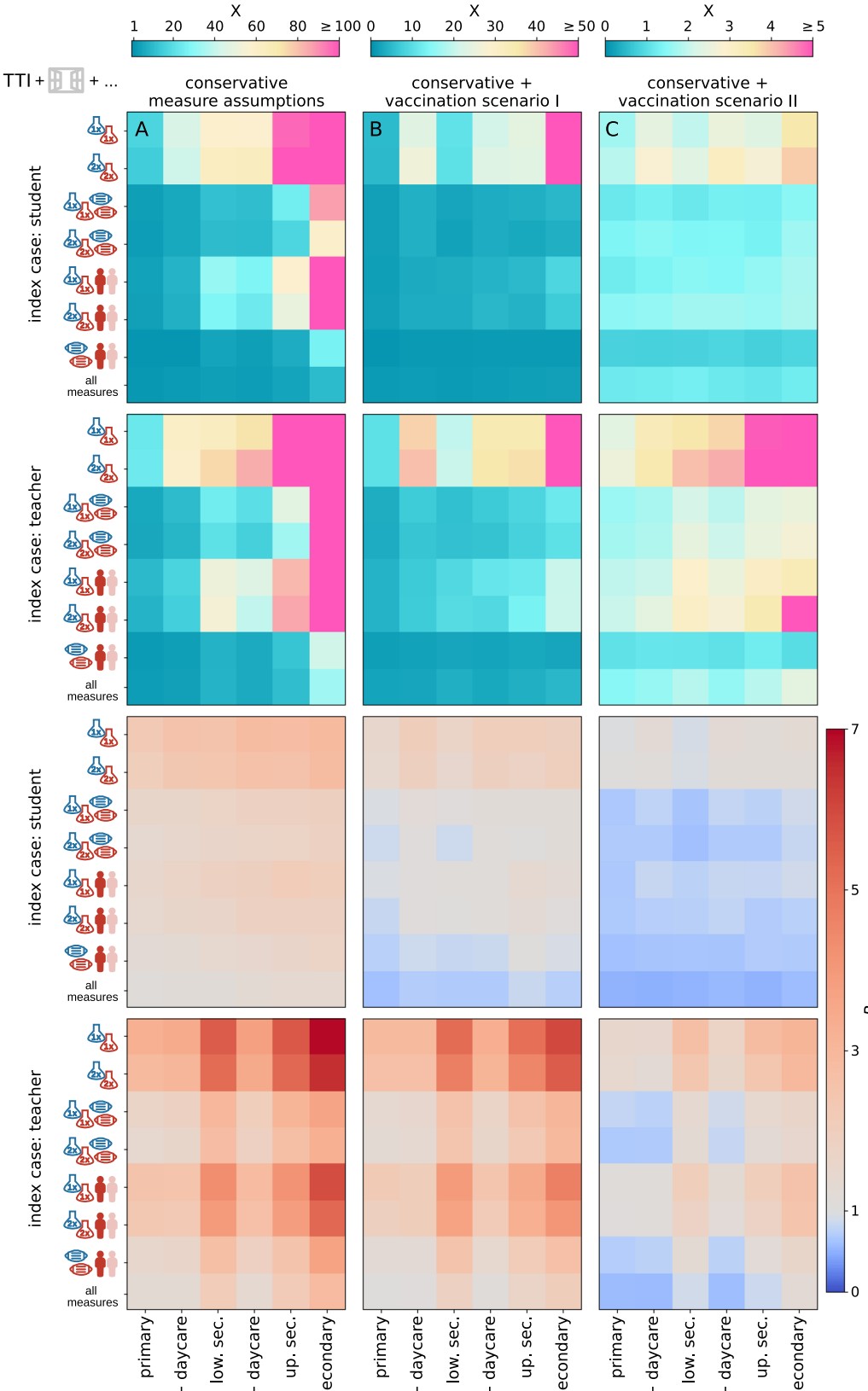

**Fig. 5 Fold increase of the average cluster size with respect to the results shown in Fig. 4, *X*, in the first two rows (note the different colour bars).** The bottom two rows show the respective *R*. Column (**A**) shows the situation if conservative assumptions are made for the effectiveness of the various implemented measures. Column (**B**) shows a scenario with conservative assumptions of measure effectiveness where 80% of teachers, 60% of family members and 0% of students are vaccinated. Column (**C**) shows a scenario with conservative assumptions in which an additional 50% of students are vaccinated. Results are shown for student source cases (rows one and three) and teacher source cases (rows two and four) for each measure package (row within each tile) and school type (column within each tile).

yields that the risk for transmission decreases by about 6% [0%; 27%] upon contact with a six year old person compared to a contact with an 18 year old person. Further, and according to our model much more importantly, secondary schools have contact networks of a completely different structure (see Fig. 2 for a comparison between a primary and a secondary school). In Austria, the average secondary school has 28 classes with 24 students each and a total of 70 teachers, whereas primary schools have 8 classes with 19 students each and 16 teachers in total. Typically teachers in secondary schools often change between class rooms during a day, whereas primary school teachers supervise only one class. Hence, contact networks in secondary schools are both denser and larger than in primary schools, which together with the age dependence of the transmission rates, leads to the observed differences in cluster sizes.

In line with the cluster data, we also find that clusters with teachers as the source result, on average, in a larger number of successive cases than clusters originated from students. There are multiple reasons for this increased transmission risk for infected teachers. First, teachers have to speak loudly facing all students for a substantial amount of time. Second, particularly in primary schools they contribute more to the spread due the age dependence of the transmission risk. Third, particularly in secondary schools they have a higher degree in the contact network, since teachers visit more classes per day as compared to primary schools. Mitigation measures that target teachers are therefore a necessary prerequisite to control the spread of SARS-CoV-2 in schools.

Here we analyzed four types of mitigation measures, namely (i) requirements to wear face masks during lessons when students sit at their desk, (ii) room ventilation, (iii) class size reduction, and (iv) screening for SARS-CoV-2 infection in asymptomatic students and teachers by means of antigen testing. We find that each of these measures by itself contributes to curbing the virus spread, but particularly in secondary schools or in schools with day care, large clusters are still likely to occur regularly, unless several measures are combined. The most effective measure in terms of reducing cluster size is frequent preventive testing of teachers and students using antigen tests with a fast result turnover, followed by class size reductions, mask-wearing and room ventilation. In primary schools, in general it is necessary to combine at least two of these measures to reduce the reproduction number below 1, whereas in (lower or upper) secondary schools it is necessary to combine at least three measures.

The effectiveness of each measure in terms of cluster size reduction depends on how well it can be implemented in practice. We found that linear decreases in measures' efficiency (e.g., participation ratios in class size reductions or testing) translate into exponential increases in cluster sizes. Therefore, a more stringently and consistently implemented measure might outperform even a measure that would be more effective under ideal circumstances. Means and incentives to ensure the proper implementation of each measure in practice are the key to success. For instance, a negative SARS-CoV-2-test result could be made mandatory to be allowed to attend school, or class rooms could be equipped with $CO_2$ sensors to ensure a proper room ventilation regime. Such "enforced" measures are to be preferred with respect to theoretically more effective measures that cannot be controlled, e.g., class size reductions if a substantial part of the students still visits the school (or a care facility) on each day of the week due to work obligations of the parents.

In a "worst case" scenario where measure effectiveness is assumed to be low, no combination of measures is sufficient to achieve control of the virus ($R < 1$) for all school types. In such a scenario, vaccination of a significant share of teachers and household members is necessary. Once larger parts of the student population are vaccinated, a combination of two measures for primary and lower secondary schools and three measures for upper secondary and secondary schools is again sufficient to contain the spread of the virus, even if each measure is not implemented in an optimal way.

Our work is subject to several limitations, pertaining both to the observational cluster data our calibration relies on, as well as to simplifications employed in the model.

The issue of age-dependent transmission risk is currently controversially discussed (see for example Anastassopoulou et al. 2020[39]). The transmission dynamics can be attributed to information bias, due to a higher share of asymptomatic cases in children – or to a mix of information bias and true decreased transmission risk among children. Our data clearly confirms that the probability for asymptomatic courses of the infection decreases with age (see Fig. 8). TTI strategies are typically triggered by the occurrence of a symptomatic case. Cluster cases might have been missed because not all contacts of the cases have been tested for infection. Since adults with a SARS-CoV-2 infection are more likely to develop symptoms and therefore to be tested, source cases might have been identified over-proportionally among teachers than among students. This might result in missed instances of student-to-student and student-to-teacher transmissions. The data on which our calibration is based upon represents a time at which the social environment of confirmed positive cases was stringently tested by the Austrian authorities. Specifically, authorities tested both category 1 and category 2 contact persons without discriminating between symptomatic and asymptomatic presentation. Nevertheless, we recognize that due to an increasing strain on the testing and tracing resources in some periods, this protocol might have been modified to some extent to preferentially test symptomatic cases. The (small) decrease of transmission risk for younger children that results from our calibration could be influenced by such observational biases.

The agent-based model makes several coarse-graining assumptions. First, to limit computational cost, we chose a time resolution for the simulation of 1 day. This can lead to a simplified representation of the infection dynamics, especially in the early days of an infection, when viral load increases approximately exponentially. Nevertheless, since epidemiological parameters controlling infection dynamics are themselves drawn from distributions, the coarse time resolution should not lead to any artifacts, for example in interaction with timed measures (weekly screening).

Second, we limit possible contacts between agents in the model to a set of the most frequent interactions arising in the school context between the predominant types of agents. We ignore other interactions and agent types, such as janitors. These interactions are assumed to be less frequent and less relevant than the interactions represented in the model. Furthermore, there is no available data on these contact types, which would render modelling them to guesswork.

Third, this analysis is based on cluster tracing data and educational statistics from Austria and therefore it remains to be seen how well our findings generalise to other countries. Class sizes and student-to-teacher ratios in Austrian schools are close to the EU average[40] which suggests that similar school-type-specific contact networks as used in our study might also be relevant for other European countries.

Fourth, we base many of our parameter choices, especially the estimates of the effectiveness of NPIs, on preliminary literature. With the sensitivity analysis of measure effectiveness included in this study, we try to cover different plausible scenarios. Nevertheless these estimates remain a source of uncertainty in the model.

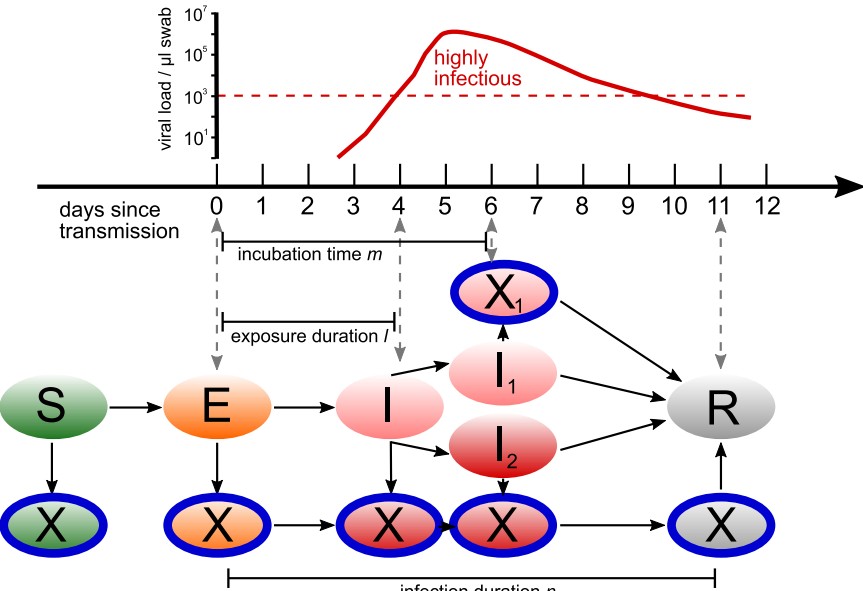

**Fig. 6 Agents in the epidemiological model can be in the states (ellipses) susceptible (S), exposed (E), infectious (I), infectious without symptoms (I1), infectious with symptoms (I2) and recovered (R).** Possible state transitions are shown by arrows. In each of these states, agents can also be quarantined (X), preventing them from interacting with other agents. Transitions between states follow the development of the viral load in the host sketched above.

Fifth and finally, regarding the effectiveness of vaccinations, we assume a constant effectiveness that is the same for every agent, and a static ratio of vaccinated agents. However, both a potential waning immunity and increased vaccine uptake can be expected to occur on time scales that are much longer than the time scales on which the model dynamics unfolds.

In conclusion, we find that different types of schools require different combinations of preventive measures. The ideal mix of mitigation measures needs to be more stringent in secondary schools than in primary schools, and needs to preferentially focus on teachers as sources of infection. Even under strict prevention measures, larger clusters in schools will still occur at regular intervals when the incidence in the general population is high enough. However, in this work we have shown that when keeping schools open during the COVID-19 pandemic a calculable risk can be achieved by a combination of stringently enforced measures which can be successively relaxed as larger portions of the population become vaccinated.

## Methods

Due to the secondary use of de-identified data on school outbreaks, the study was exempt from ethical approval by the relevant IRB. To perform the interviews in this study, we acquired informed consent by interviewees.

**Empirical observations of SARS-CoV-2 clusters in Austrian schools.** Clusters of SARS-CoV-2 cases among Austrian residents are identified at the Agency for Health and Food Safety (AGES) in cooperation with the responsible regional public health authorities. A cluster is defined as a group of at least two cases of a confirmed SARS-CoV-2 infection, which are epidemiologically linked by means of an infector and infectee (i.e. successive case). A "school cluster" includes at least one case generated by in-school transmission. The source of a school cluster introduces the virus into the school setting, is either a teacher-case or a student-case, and is generated by out-school transmission, in settings such as household, work place, leisure activity, or in an unknown setting (referred to as teacher-source case or student-source case throughout the entire text, could be denoted as index case elsewhere).

As of December 22nd 2020, we identified 616 clusters with at least one school transmission with a starting date between calendar week 36 and 45, 2020. The starting date of a school cluster is defined as the date of laboratory diagnosis of its source case. The 616 clusters involved 9232 cases. Out of these, 3498 were school cluster cases, including 2822 student-cases and 676 teacher-cases. Out of the school cluster cases, 464 were source cases and 3034 cases were generated by in-school

transmission. Each cluster was assigned to one of the school-types primary, lower secondary, upper secondary, secondary and inconclusive. Data on the exact school type was not available at the time of our analysis, therefore we assigned the clusters based on the age of students using the following algorithm:

1. primary: the age of all affected students is ≤ 10 years.
2. lower secondary: the age of all affected students is within the interval [10, 15] years.
3. upper secondary: the age of all affected students is ≥ 15 years.
4. secondary: the age of all affected students is ≥ 10 years.
5. inconclusive: otherwise.

In total, 286 cases were related to primary schools (69% students), 762 to lower secondary schools (79% students), 388 to the upper secondary schools (89% students) and 810 to the secondary schools (88% students), see also Fig. 1B. The share of student-source cases was lowest in primary schools (6%), followed by lower secondary (43%), secondary (64%) and upper secondary (82%), see Fig. 1A. The clinical presentation was clearly age-dependent. While 4 out of 6 students younger than 6 years old were asymptomatic, the proportion of asymptomatic cases dropped by increasing age from 61%, 49%, 33% and 16% for the age groups 6–10, 11–14, 15–18 and adults (including students and teachers), respectively. Figure 1C shows the distribution of cluster size among different school types. Overall, the amount of clusters with size 2, 3–9, 10–19 and 20+ was 40%, 49%, 8% and 3%, respectively.

**Agent-based simulation**
*Model.* We simulate the infection dynamics in schools using an agent-based model[41]. The model includes three types of agents: students, teachers, and their respective household members. The model couples in-host viral dynamics with population dynamics. Depending on the viral load over the course of an infection, each agent is in one of five states: susceptible (S), exposed (E), infectious (I), recovered (R) or quarantined (X) (see Fig. 6). In addition, after the presymptomatic phase, agents can stay asymptomatic (I1) or develop symptoms (I2). Age also influences the transmission risk (see Section "Transmission risk" below). Agents remain in these states for variable time periods. Every agent has an individual exposure duration, $l$, incubation time (i.e., time until they may show symptoms), $m$, and infection duration, $n$ (i.e., time from exposure until an agent ceases to be infectious), as depicted in Fig. 6. For every agent, we draw values for $l$, $m$, and $n$ from previously reported distributions of these epidemiological parameters for the SARS-CoV-2 delta variant. Incubation time, $m$, is distributed according to a Weibull distribution with a mean of $4.4 \pm 1.9$ days[21]. To date, the exposure duration has not been determined for the delta variant, therefore we use the exposure duration of $5.0 \pm 1.9$[42–44] that has been reported for the original strain and subtract two days to mirror the change of the reported incubation time for delta with respect to the original strain[21]. We leave the variance that was initially reported for the original strain unchanged. Exposure duration, $l$, is therefore distributed according to a Weibull distribution with a mean of $3.0 \pm 1.9$ days. We also impose the constraint that $m \geq l$. Infection duration, $n$, is distributed according to a

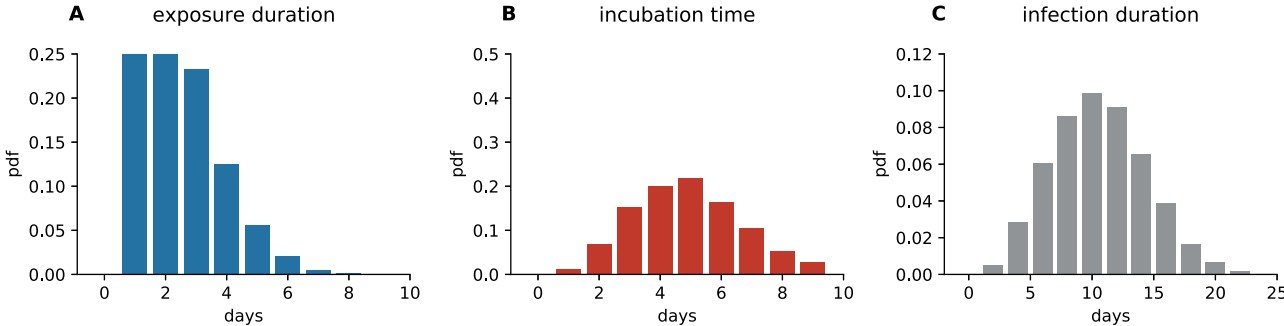

**Fig. 7 Distributions of parameters used in the agent based simulation. A** Exposure duration, $l$, (**B**) incubation time, $m$, and (**C**) infection duration, $n$.

Weibull distribution with a mean of $10.9 \pm 4.0$ days[45,46] for the original strain. Since to date there are no reports about a change in infection duration, we keep the infection duration that was reported for the original strain also for simulations with the delta variant. We impose the additional constraint of $n > l$. Distributions for $l$, $m$ and $n$ are shown in Fig. 7. Infections are introduced to the school setting through a single source case that can either be a student or a teacher. The source case starts in the exposed state on day 0 of the simulation which corresponds to a randomly chosen day of the week. All other agents start in the susceptible state.

Agents interact by means of networks of contacts specific to the school setting and the day of the week (see Section "Contact networks" below). At every step (day) of the simulation, agents interact with other agents in the neighborhood of their day-specific contact network. Infected agents can transmit an infection to susceptible agents, unless one of them is quarantined. Quarantined agents are represented by isolated nodes in the contact network.

*Transmission risk.* During every interaction, an infected agent can transmit the infection to the agents they are in contact with (specified by the contact network, see Section "Contact networks" below). Transmission is modelled as a Bernoulli trial with a probability of success, $p$. This probability is modified by several intervention measures and biological mechanisms $q_i$, where $i$ labels the measure or mechanism. Here, we consider nine such mechanisms (see Table 1 for details). This includes the modification of the transmission risk due to the type of contact between agents (represented by $q_1$), due to the age of the transmitting and receiving agents ($q_2$ and $q_3$, respectively), the infection progression ($q_4$), having or not having symptoms ($q_5$), mask wearing of the transmitting and receiving agent ($q_6$ and $q_7$, respectively), room ventilation ($q_8$), and immunization ($q_9$). Therefore, the probability of a successful transmission is given by the base transmission risk, $\beta$, of a household contact modified by the combined effect of these nine intervention measures or biological mechanisms,

$$p = 1 - \left(1 - \beta \prod_{i=1}^{9}(1 - q_i)\right).$$

The base transmission risk $\beta$ is calibrated to reflect the reported transmission risk in households (see Section "Calibration" below).

To model the reduction of transmission risk due to the type of contact between agents, we classify contacts into two categories. "Household" contacts are characterised by very long and physically very close interactions and occur only between members of the same household. "School" contacts are contacts that occur in the school context and are of a lower intensity (duration, physical closeness) than household contacts. We model the reduction of the transmission risk, $q_1$, for school contacts as compared to household, which is calibrated given empirical data of outbreaks observed in Austrian schools (see Section "Empirical observations of SARS-CoV-2 clusters in Austrian schools" above).

Infection dynamics of SARS-CoV-2 differs between children and adults[13,14]. There is still uncertainty concerning how individual biological or epidemiological factors impact the transmission of and susceptibility to an infection with SARS-CoV-2 in children. Susceptibility in children is believed to be inhibited due to a lower number of ACE2 receptors[47] that are necessary for the virus to enter cells. Transmission is believed to be inhibited due to the lower number of symptomatic cases in children[13] and their smaller lung volumes, which reduces the amount of virus-laden aerosols emitted by infected children[48]. Yet, an accurate quantification of these effects still eludes us. In addition, children of different ages might express different contact characteristics (duration, proximity), which in turn would also introduce an age-dependence of the transmission risk. We model all these possible influences as a compound factor that introduces an age-dependence of the transmission risk. In addition and to reduce the number of parameters that need to be calibrated we assume that both, reduced transmission and reduced susceptibility in children, contribute equally. We model both effects as a linear decrease of risk depending on the age of the student. The modification of the transmission risk due to the age of the transmitting agent $q_2$ is modelled as a linear decrease of the

infection risk with every year an agent is younger than 18,

$$q_2(y_{\text{transmit}}) = \begin{cases} c_{\text{age}}|y_{\text{transmit}} - 18| & \text{if } y_{\text{transmit}} \le 18 \\ 0 & \text{else,} \end{cases} \quad (1)$$

where $y_{\text{transmit}}$ is the age of the transmitting agent and $c_{\text{age}}$ is the slope of the linear relationship, which is calibrated using empirical observations of clusters in the school setting (see Section "Calibration" below). The modification of the transmission risk due to the age of the contracting agent $q_3$ is also modelled as a linear decrease of the infection risk with every year an agent is younger than 18, using the same slope $c_{\text{age}}$,

$$q_3(y_{\text{contract}}) = \begin{cases} c_{\text{age}}|y_{\text{contract}} - 18| & \text{if } y_{\text{contract}} \le 18 \\ 0 & \text{else,} \end{cases} \quad (2)$$

where $y_{\text{contract}}$ is the age of the contracting agent.

The modification of transmission risk due to a changing viral load over the course of an infection, $q_4$, is modelled as a trapezoid function that depends on the time an agent has already been exposed to the virus, $t$, given the exposure duration, $l$, incubation time, $m$, and infection duration, $n$, of the infected agent:

$$q_4(t) = \begin{cases} 0 & \text{if } l < t \le m \\ 1 - \frac{t-m}{n-m+1} & \text{if } t > m \text{ and } t \le n \\ 1 & \text{else.} \end{cases} \quad (3)$$

This means that the transmission risk is constant and high during the first few days after the exposure phase and until symptoms occur (in symptomatic agents), and then decreases linearly until the end of infectiousness is reached. This development is in line with investigations of the development of the viral load in patients infected with the original SARS-CoV-2 strain[45,49].

The reduction of the transmission risk due to not having symptoms $q_5$[50], wearing a mask $q_6$ and $q_7$[51], and ventilating the room $q_8$[37], is modelled using literature values for the respective effects (see Table 1) and is in line with an overall transmission risk reduction of around 80% reported in a recent review on mask effectiveness.

*Vaccinations.* To model different vaccination scenarios in schools, the model allows for the specification of a ratio of vaccinated agents per agent type. If, for example, 50% of students in a given school are vaccinated, we pick 50% out of all students at random at model initialisation and assign them the status "vaccinated".

Being vaccinated reduces an agent's chance of getting infected. Specifically, we model the effectiveness of the BNT162b2 and ChAdOx1 vaccines against high viral load 90 days after the second dose of vaccination, i.e. including a drop in effectiveness with time. Pouwels et al.[52] report a 78% and 61% effectiveness against high viral load of BNT162b2 and ChAdOx1, respectively. As our population will have a mix of different vaccines and times passed since the second dose, and effectiveness is expected to drop further as time passes, we chose a conservative estimate of vaccine effectiveness of 60% in our simulations. Therefore, if a vaccinated agent comes in contact with an infected agent, an additional reduction of transmission risk of 0.6 ($q_9$) applies.

Since the viral load of vaccinated people infected with the delta variant seems to be similarly high as the load of unvaccinated people[53], we do not assume a lower infectiousness of vaccinated people if they get infected as a conservative assumption.

Vaccination status does not change throughout the simulation but vaccinated agents that are infected will recover at the end of their infection and will subsequently be completely immune to reinfection after recovery.

*Testing and tracing.* In all simulations, upon first developing symptoms, agents are immediately quarantined and tested with a PCR test that has a one-day result turnover time in the calibration scenario and a two-day result turnover time in every other scenario. This reflects the situation that during the time period from which our calibration data stems, testing and tracing was still sufficiently functional, while with increasing case numbers in late autumn 2020, the testing and

**Table 3 Contacts that arise from different situations in the school context between different agent groups.**

| Situation | Group 1 | Group 2 | Strength |
|---|---|---|---|
| Student household | Student | Household member | Household |
| Siblings in student household | Student | Student | Household |
| Teacher household | Teacher | Household member | Household |
| Student or teacher household | Household member | Household member | Household |
| Classmates | Student | Student | School |
| Daycare group | Student | Student | School |
| Friends | Student | Student | School |
| Meeting | Teacher | Teacher | School |
| Daycare supervision | Teacher | Teacher | School |
| Team teaching | Teacher | Teacher | School |
| Teaching | Teacher | Student | School |
| Daycare supervision | Teacher | Student | School |

Contacts are always bi-directional. Every contact is qualified by a contact strength ("household" or "school"), that condenses both the typical duration and physical proximity of participating parties.

tracing capacity reached its limits and delays increased. We gathered this information on typical test turnover times in our stakeholder interviews, described in Section "Interviews with school personnel" below. In addition to testing of symptomatic agents (diagnostic testing), during the calibration period (autumn 2020) a positive diagnostic test triggered a test of all teachers and students in the school (background screen). This policy has since changed and positive tests do not trigger a background screen in schools anymore. We therefore implement background screens only in the calibration simulations and not in all other simulations. Diagnostic testing and quarantining of contact persons (test-trace-isolate, TTI) occur in every scenario regardless of calibration period or not, and even if no additional measures are implemented.

An additional preventive measure is screening testing, which intends to identify infected people who are asymptomatic and do not have known, suspected, or reported exposure to SARS-CoV-2. Screening tests can be performed in defined intervals. When screening tests occur once per week, it is performed every Monday, when twice a week then every Monday and Thursday. These tests have a same-day result turnover, reflecting the fact that results are usually available within minutes after the test.

The sensitivity of AG tests depends on the viral load of the swabs[25,54]. Therefore, the sensitivity of AG tests might depend on both, the clinical presentation of the disease, as well as the number of days a patient has been infected at the time of testing. The Austrian Agency for Health and Food Safety (AGES) validated the performance of AG test with the anterior nasal sampling (NS) and found a sensitivity of 40.7% among asymptomatic infected persons and of 75.9% in mildly symptomatic infected for the original strain[55]. The high sensitivity for mildly symptomatic patients is consistent with other studies that report sensitivity for symptomatic patients[56,57].

To account for the dependence of the sensitivity of AG tests on the viral load of the patients, we approximate the sensitivity by a step-function: in our model, the sensitivity of AG tests is 1.0 during a restricted time window between 6 and 11 days after exposure and 0 otherwise. We do not differentiate between agents with symptomatic and asymptomatic courses. Under the assumption that the likelihood of being tested within the scope of screening by use of AG tests is uniformly distributed over the entire course of the infection and with a mean infection duration of 10.9 days[45,46], this results in an overall sensitivity of AG tests of 0.45 in our model. Therefore, the AG test sensitivity used in our model lies in between the sensitivities that were reported for asymptomatic (40.7%) and mildly symptomatic (75.9%) patients for the AG tests used in the school context in Austria[55].

PCR tests are very sensitive and require low numbers of virus copies in clinical samples to successfully detect an infection with SARS-CoV-2[25]. We model this by allowing PCR tests to detect infections starting four days after exposure and up until 11 days after exposure.

If an agent receives a positive test result (after the specified result turnover time of the respective test technology), their category 1 contacts[58] are traced and quarantined. In our simulations, for the calibration, category 1 contacts were those with household members and members of the same class, i.e. if a teacher tests positive, all classes that were taught by the respective teacher are quarantined and if a student tests positive, their class is also quarantined. This reflects the practice of Austrian schools to trace and quarantine these types of contacts in the time period used for calibration. For other simulation runs (i.e., non-calibration), category 1 contacts were defined as only those with household contacts to the infected agent. This reflects the changed practice in Austrian schools after the re-opening of schools in February 2021 with a strict FFP2 mask mandate.

Tracing is considered to occur instantly and contact persons are quarantined without time delay, as soon as a positive test result returns. Quarantined agents will stay in quarantine for 10 days, corresponding to the recommendations of the Austrian authorities regarding the quarantine of contact persons[58]. Quarantined agents stay in quarantine, even if they receive a negative test result during that time.

*Contact networks.* The contact networks for schools are modelled to reflect typical social interaction structures in Austrian schools (see Supplementary Information section 1 for details) and informed by interviews with school personnel to gather information on the daily life in Austrian schools during the pandemic (see Section "Interviews with school personnel" below, and Supplementary Information Section 4). Schools are defined by the number of classes they have and the school type, which determines the age structure of their students. For every school type, we use the average number of classes and the average number of students per class as reported in the most recent Austrian school statistics[33]. The number of teachers in a given school is determined based on the number of classes and the school type, given the number of teachers per class listed in the Austrian school statistics[33]. Adjustments for daycare and extra language teachers in certain school types were made as they were pointed out to us in the stakeholder interviews (see also Supplementary Information Section 1). The Austrian school statistics do not differentiate between schools with and without daycare. Therefore we assume that schools with and without daycare are not significantly different in the number of classes and the number of students per class. Nevertheless, the number of teachers in schools with daycare is slightly higher in primary and lower secondary schools. See Table 2 for the number of classes, students and teachers for every school type modelled in this work. Every student and teacher has a number of family members drawn from distributions of household sizes and a number of children corresponding to Austrian households[59]. For students we additionally impose the condition on the distributions that each household has at least one child. If a student household has a second child (sibling) that is eligible (by age) to attend the same school, the sibling will also be assigned to an appropriate class in the school, if there is still room. This practice reflects the distribution practice of children to schools in Austria, where efforts are undertaken to enable siblings to attend the same school. Students, teachers and household members are nodes in the contact network. Two exemplary contact networks are shown in Fig. 2.

Contacts (edges) between different agents in the contact network are derived from a variety of situations that create contacts between the involved agents in households and the school context, and which were derived from the stakeholder interviews (see Table 3 for a list of possible contacts). We assume that contacts that occur in households and contacts that occur in schools have different intensities. These contact intensities are calibrated using empirical data of outbreaks in schools (see Section "Calibration" below).

Contacts that arise in schools include contacts between teachers and students during lessons or daycare supervision, contacts between students of the same class during lessons and contacts between students of different classes during daycare supervision. Contacts during daycare supervision are an important factor since in practice (according to the stakeholder interviews), daycare groups are not composed of students from the same class but rather consist of an independent matching of students to groups and therefore result in the mixing of students between classes. Mixing between classes is also caused by siblings, that attend different classes but have a (household) contact.

We assume that no contacts between students occur outside households or lessons/daycare supervision, reflecting a situation in which students do not visit friends and do not have contacts to children from other classes during breaks. This assumption reflects Austrian governmental policies during the calibration period, where restrictions on private gatherings were in place. In addition, schools went to great lengths to prevent students from different classes from meeting, introducing staggered beginning and ending times and movement restrictions in school buildings. However, to study situations without such restrictions we introduce a number of additional "friendship" contacts for our sensitivity analysis and "conservative" scenario. Friendship contacts are introduced by randomly picking a number of students from every class, corresponding to a desired ratio, and by introducing additional edges to randomly selected students from other classes. In

our sensitivity analysis, we vary the ratio of students that have friendships outside their own class between 0% and 40%.

Next to contacts among students and between students and teachers, teachers can have a variety of contacts with each other. Next to teaching the same class together with another teacher (team-teaching) and shared daycare supervision duties, teachers regularly meet their colleagues to discuss lessons. Literature on these social networks among teachers is scarce but there is a study[60] that puts the network density score of to "engage in conversation regularly" between teachers at 0.25 and of to "socialize with outside school" at 0.06, which is consistent with the evidence gathered during the stakeholder interviews. To model these contacts, teachers that teach or supervise together will have a contact. In addition, we randomly create a number of connections between teachers that reflects the network density scores reported by Hawe et al.[60].

Contacts in our contact network are dynamic and do not exist on every day of the week, reflecting the periodic organization of teaching in Austria. Students are usually at school from Mondays through Fridays and are at home on Saturdays and Sundays. To reflect this, on weekends only household contacts exist between agents. To implement the prevention measure of reduced class sizes, we remove all school-related contacts for a fraction of the students of every class on every second day.

**Calibration and parameter choices.** Some model parameters can be taken over from the existing literature (see Section "Transmission risk" above) or empirical observations of characteristics of infection spread in the school context in Austria (see Section "Empirical observations of SARS-CoV-2 clusters in Austrian schools" above). For our model, that leaves a total of three free parameters that have to be calibrated to reproduce the observed dynamics of infection spread as closely as possible: (i) the base transmission risk of a household contact, $\beta$, (ii) the reduction of transmission risk of contacts in the school context as compared to a household contact, $c_{contact}$, and (iii) the linear age dependence of transmission risk and susceptibility, which we consider to have the same slope $c_{age}$ and an intercept of 1 for agents aged 18 or older.

*Household contacts.* The cumulative risk of adult members of the same household to get infected over the course of the infection of an infected household member was estimated as 37.8% [61] for the original strain of the virus. We calibrate the base transmission risk, $\beta$, between adult agents in our model such that it reflects this cumulative transmission risk and then multiply it by a factor of 2.25 in our subsequent simulations, to reflect the increased transmissibility of the delta variant[19,20]. For household transmissions between adults, the only relevant factors that modify the base transmission risk are the reduction due to the progression of the disease, $q_4(t)$, and the reduction in case of an asymptomatic course, $q_5$. The values for both of these factors are taken from the literature[45,49,50]. Therefore, for one contact on day $t$ after the exposure, the probability of a successful transmission is given as

$$p(t) = 1 - \left[1 - \beta(1 - q_4(t))(1 - q_5)\right].$$

In our model, we draw the relevant epidemiological parameters (exposure duration, infection duration, symptomatic course) from corresponding distributions[21,42–46] individually for every agent. To calibrate $\beta$, we create pairs of agents and let one of them be infected. We then simulate the full course of the infection (from day 0 to the end of the infection duration $n$) and perform a Bernoulli trial for the infection with a probability of success $p(t)$ on every day $t$. We minimize the difference between the expected number of successful infections (37.8%) and the simulated number of successful infections by varying $\beta$. This results in an optimal value of $\beta = 0.0737 \pm 0.002$ (mean $\pm$ SD) or an average risk of 7.4% per day for a household member to become infected. We note that the reduction of transmission risk and susceptibility due to the age of the transmitting and receiving agents is treated and calibrated separately. This is why we only calibrate the transmission risk between adults here and calibrate the age dependence of transmission risk and susceptibility $c_{age}$ separately.

*Matching cluster sizes and group distributions.* For the calibration of the other two free parameters $c_{contact}$ and $c_{age}$, we compare the distribution of cluster sizes and the distribution of the number of infected agents across the agent groups "student" and "teacher" between our simulation and empirically observed clusters in Austrian schools (see Section "Empirical observations of SARS-CoV-2 clusters in Austrian schools" above).

During calibration, for the other simulation parameters we use settings that most closely match the situation in Austrian schools in the time period from which the empirical observations were taken (weeks 35–46, 2020): Source cases are drawn from the empirically observed distribution of source cases between teachers and students (see Fig. 1A). The age dependence of the probability of developing a symptomatic course of infection is matched to the empirically observed age-dependence (see Fig. 8). Only diagnostic testing with PCR tests with a one-day turnover was in place, followed by a background screen (test of all teachers and students in the school with PCR tests) in case of a positive result. There were no preventive screens and no follow-up tests after a background screen. Contacts between students and teachers in the same class as the infected person were considered to be "category 1" contacts[58] and contact persons were quarantined for 10 days and remained isolated for the full quarantine duration, even in case of a

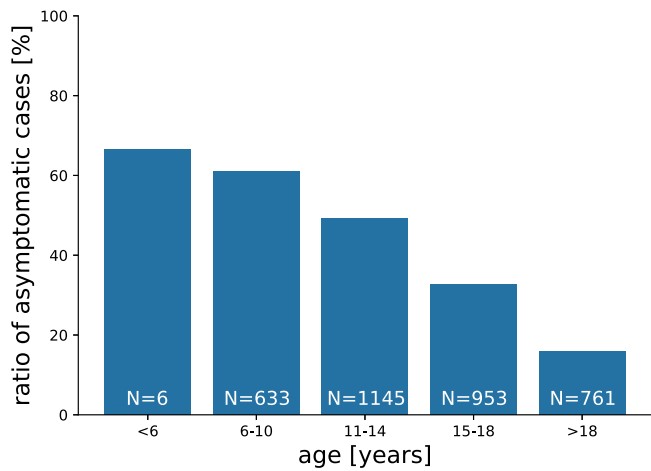

**Fig. 8 Ratio of asymptomatic cases by age group.** Age groups 6–10 years, 11–14 years and 15–18 years correspond to the usual age of students in primary, lower secondary, and upper secondary schools in Austria.

negative test result during isolation. Teachers and students did not regularly wear masks during lessons. Teachers and students did wear masks in hallways and shared community areas and contacts between students of different classes were avoided. All students of a class were present every day i.e., no reduction of class sizes was employed.

For the calibration, we match the cluster characteristics of our simulations with the cluster characteristics of empirically observed clusters in Austrian schools. Specifically, we optimize the sum of the squared differences between the empirically observed cluster size distribution and the cluster size distribution from simulations $e_1$, and the sum of squared differences between the empirically observed ratio of infected students versus infected teachers, and the simulated ratio. Using the settings for prevention measures described above to find optimal values for the two remaining free parameters, we first simulate parameters from a coarse parameter grid spanned by the following ranges ([start:stop:step]): $c_{contact}$: [0:1:0.05] and $c_{age}$: [−0.09:0.0:0.01], conducting 500 simulations per parameter combination and school type. Since the empirical data we compare our simulation results to does not differentiate between schools with and without daycare of a given school type, we assume that 50% of the schools of a given school type are schools with daycare. This approximates the percentage of schools with daycare in Austria[62]. We therefore simulate ensembles for primary schools, primary schools with daycare, lower secondary schools, lower secondary schools with daycare, upper secondary schools and secondary schools. We calculate the overall difference between the simulated and empirically observed cluster characteristics as

$$E = \sum_i \frac{e_{1,i} + e_{2,i}}{N_i},$$

where $N_i$ is the number of empirically observed clusters for school type $i$. After we identify the parameter combination that minimizes $E$ in the coarse grid search, we perform a refined grid search around the current optimal parameter combination for the ranges $c_{contact}$: [0.23:0.35:0.01] and $c_{age}$ [−0.03:0.0:0.0025], conducting 4000 simulations per parameter combination. We then repeat the optimization process as described above.

To assess the uncertainty associated with the optimal parameter combination we implement a bootstrapping approach on the simulated ensembles: We randomly (with replacement) sample 2,000 out of the 4,000 simulation runs for every parameter combination, calculate the difference $E$ between the empirically measured and simulated (subsampled) outbreak size distributions and determine the parameter combination that minimizes $E$ for a total of 1,000 such samplings. We use the median values of $c_{contact} = 0.30$ and $c_{age} = −0.005$ as calibrated values for all subsequent simulations. This means that contacts in the school context have a transmission risk that is reduced by 70% as compared to household contacts. Children that are younger than 18 years have a transmission risk that is reduced by $0.005(18 − y)\%$, where $y$ is the age. Therefore, the transmission risk and susceptibility of a six year old child is 6% lower than that of an 18 year old. The sensitivity analysis for changes in $c_{contact}$ and $c_{age}$ is performed using the [0.025; 0.975] percentile values from the bootstrapped distributions for $c_{contact}$ and $c_{age}$, which are [0.26; 0.34] and [−0.0225; 0], respectively.

To ensure that the thus determined optimal parameter combination is not subject to large changes if different distance measures are used to calculate $E$, we introduce additional distance measures between the empirical and simulated distributions of outbreak sizes: in addition to the sum of squared differences, we calculate the $\chi^2$ distance, the Bhattacharyya distance, the Pearson and Spearman correlation (we use the absolute value of the difference between 1 and the correlation coefficient as distance measure), and the slopes of pp- and qq-plots

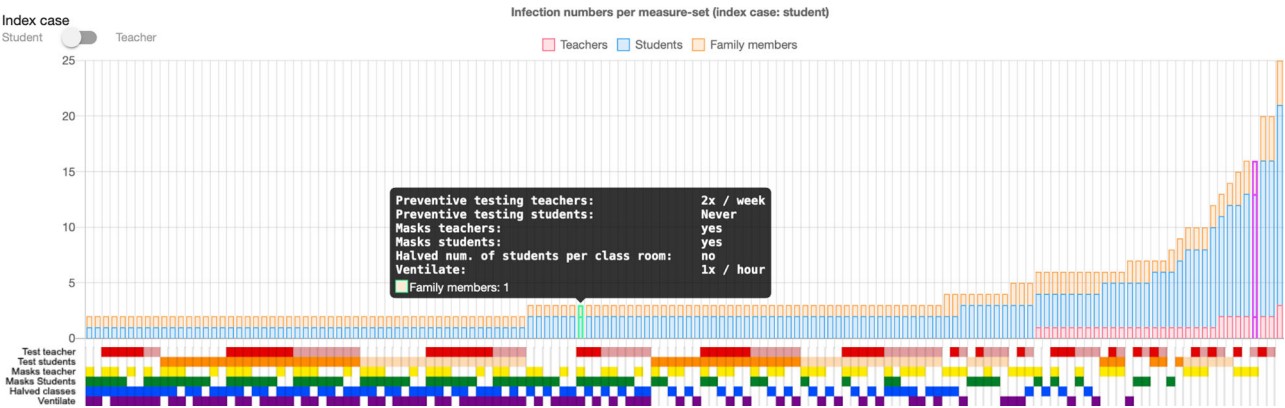

**Fig. 9 Overview of the online interactive simulation viewer displaying the effects of measure combinations on cluster size.** Each bar represents one set of measures. Bars are separated into three stacks, representing the number of infected students, teachers, and family members. The bar representing the currently selected set of measures is highlighted in purple (right). A popup window displays the measure configuration of a bar on mouse-over. Below the chart, six rows of coloured squares visually encode the set of measures that yielded the respective simulation result. A row (representing a measure) can be clicked, in order to re-sort the plot by the selected measure and thus enable a better comparison of the measure's effects. The button on the top left corner allows a user to switch the results shown in the plot between simulations that had a student or a teacher as a source case.

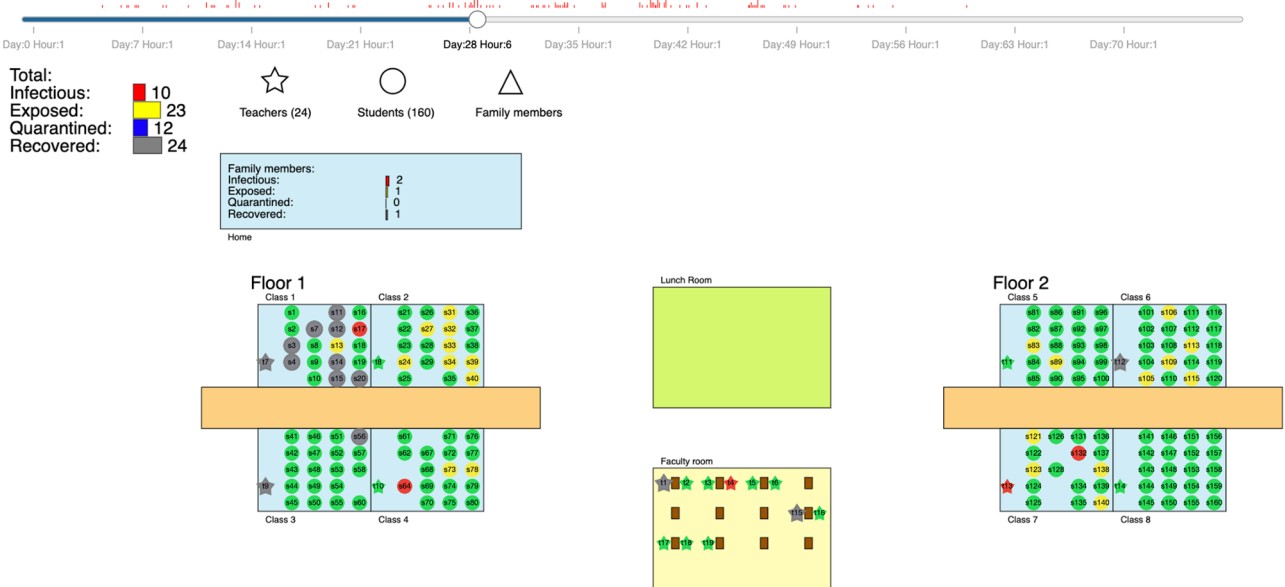

**Fig. 10 The "outbreak simulation view" in the online visualization displays the schedule of students and teachers over time, illustrated in a schematic view of the respective school.** Agents (nodes) change their colour according to their current infection state: susceptible (green), exposed (yellow), infectious (red), recovered (grey). Red bars on the timeline (top) indicate when infections occur. Outside of school-hours, agents spend their time at home (indicated by a single box on the top left) where infections with family members can occur. Family members are hidden from the animation, but their state is tracked as well, and displayed as a histogram on top of the home. A summary of the state of all agents is displayed on the top left corner below the time line.

(we use the absolute value of the difference between 1 and the slope as distance measure). Heatmaps of values for the different distance measures and different parameter combinations are shown in Supplementary Fig. 1. The median values and [0.025; 0.975] confidence intervals for all seven distance measures are shown in Supplementary Fig. 2. While there is some variation between the different distance measures, all confidence intervals overlap and we confirm that the optimal parameter choice only slightly depends on the chosen distance measure.

We note that in our calibrated model, a significant number of simulations do not lead to further transmission within the school (63% for primary schools and primary schools with daycare, 44% for lower secondary schools, 55% for lower secondary schools with daycare, 44% for upper secondary schools and 40% for secondary schools). As the cluster tracing data does not contain such events, we cannot validate the number of these instances.

*Ventilation.* Using the COVID-19 transmission risk calculator[37,63], we calculate the ventilation efficiency of a short and intensive ventilation of the classroom once per hour during one lesson for the teacher and student source case.

According to the building regulation for schools in Austria[64], classrooms must have a minimum area of 1.6 m² / student and a total minimum area of 50² in primary and lower secondary schools. Since the maximum class size we simulate (secondary schools) is 30 students (30*1.6 m² = 48 m²), we can assume all classrooms have a size of ~50 m². The individual infection risk is independent of the number of people in the room. Mask wearing linearly reduces infection risk and is therefore modelled as a separate parameter that influences the infection risk.

To calculate the reduction of transmission risk for one ventilation / hour, we therefore use the following parameters for the teacher source case:

- Speaking volume: 3 (loud talking is assumed during teaching)
- Mask filter efficiency (exhale): 0
- Mask filter efficiency (inhale): 0
- Ratio of time speaking: 50% (teacher teaching a class)
- Breathing volume [l/min]: 10 (adult)
- Room area 50 m² (building regulations[64])
- Room height 3.2 m (building regulations[64])

- Duration: 1 hour (one lesson)
- Air exchange rate: 2 (corresponding to one ventilation / hour)

This results in an an individual infection risk of 1.9% per person in the room. Compared to the individual infection risk if of 5.3% for no room ventilation (air exchange rate of 0), this is a reduction of 64%. A second simulation tool yields similar results (see https://www.corona-rechner.at/, accessed September 14, 2021).

For the student source case, we use very similar parameters, except for the speaking volume (2, normal talking), the ratio of time speaking (10%) and the breathing volume (7.5 l/min, child). This results in an individual infection risk of 0.2% if the room is ventilated once per hour and 0.55% for no ventilation. Therefore ventilation also reduced the infection risk by 64%.

**Calculating the reproduction number, _R_.** We use the reproduction number calculated from infection chains in our model to report and compare outcomes of different intervention scenarios. Since our model is agent-based, $R$ is calculated as an individual-level measure[65] by counting the number of secondary infections a focal individual has caused. Since the number of agents in our model is rather small ($\mathcal{O}(10^3)$), we expect the finite size of the model to significantly influence the number of secondary infections as the infection spreads through the system, since the pool of susceptible agents is not infinite and depletes with time. To minimize this effect, we calculate the reproduction number, $R$, of an ensemble of simulations as the average number of secondary infections caused by the source case only, disregarding the rest of the transmission chain. This approach is warranted since source cases are picked at random and no systematic biases are introduced by the local connectivity of the contact network of the source case.

While in this case $R$ is not a model control parameter that determines whether the number of infected will grow or decline, as in the classical SIR model, it is still a useful indicator to assess how likely a wide spread of the infection through the system is. If $R < 1$, for the majority of cases the source case will only infect one or no other agents. This does not exclude the possibility for rare larger clusters, but they are much less likely as for settings for which $R > 1$.

**Interviews with school personnel.** We conducted semi-structured interviews with a total of eight teachers and principals of Austrian schools (1 primary, 3 lower secondary, 1 upper secondary, and 3 secondary). The aim of the interviews was to get an impression of daily life in Austrian schools during the pandemic. The gathered information was used in three ways: (i) to design the school type specific contact networks, (ii) to design the intervention measures and (iii) to assess potential problems with the implementation of intervention measures. Interviews were conducted over video chat. The questionnaire used to guide the interviews is provided in Supplementary Information Section 4.

**Online visualization.** The online visualization aims to convey our simulation results to decision makers, as well as more general audiences, such as students and parents. Since we cannot assume our audience to be familiar with statistics or simulation, we designed our visualization in a way that allows us to guide a user through the supplied information in four sequential sections (I-IV). (I) We first provide a graphical overview of our simulation results for different measures and school types, similar to the one in Fig. 4. We then provide general information on calibration data that could be of interest to the general public, such as the infectiousness of a person during the course of their infection. (II) Users who want to explore further can then investigate detailed results for a specific school by using an input mask to configure school type and size (number of classes, and class size). The number of teachers is automatically derived from the school type and size. (III) Once a school is configured, the user is supplied with two histograms detailing the cluster size and number of quarantine days for all combinations of measures (see Fig. 9). The plots are interactive and can be re-sorted by measure in order to enable a better comparison of the individual measure effects. By default, we display the 0.9 percentile outcome of our simulations ("worst case"). Users also have the option to investigate the median and 0.1 percentile scenarios. If a user wants to see how an infection in a specific scenario (set of measures) plays out, they can select the respective histogram bar in order to switch to the cluster simulation view. (IV) The cluster simulation (see Fig. 10) shows an animation of the simulated events that led to the results of the selected set of measures for the specified school. The animation illustrates the daily school routine of students and teachers (depicted as circles) within a schematic school floor-plan. The shape of a circle indicates the agent type (student or teacher). The colour indicates the state of an agent (susceptible, exposed, infected, recovered). All sections of our visualization are accompanied by explanatory texts, in order to ensure users are informed about the functionality and the displayed information in each section. Users can switch between sections via the navigation menu on the top of the web page, and also share the results of their exploration by generating a URL that stores their school and measure selections.

**Reporting summary.** Further information on research design is available in the Nature Research Reporting Summary linked to this article.

## Data availability
The cluster tracing data used to calibrate the model in this study have been deposited at Zenodo under accession code https://doi.org/10.5281/zenodo.4706876[66]. The data includes all clusters of SARS-CoV-2 infections with at least one transmission in an educational setting recorded between calendar weeks 36 (August 31) and 45 (November 11) 2020 in Austria. Data was collected by Austrian contact tracing agencies in line with the applicable Austrian regulations. No sampling was performed and no data was excluded.

The contact networks and simulation results of this study have been deposited at OSF under accession code https://doi.org/10.17605/OSF.IO/MDE4K[67].

## Code availability
The code for the agent based simulation model is openly available under an MIT license at https://github.com/JanaLasser/agent_based_COVID_SEIRX. For the simulations in this publication, version 1.4.1 of the codebase has been used: https://github.com/JanaLasser/agent_based_COVID_SEIRX/releases/tag/v.1.4.1[41]. The code used to run and analyse simulations for schools is also openly available under an MIT license at https://github.com/JanaLasser/school_SEIRX[68]. For the simulations in this publication, version 1.0.0 has been used: https://github.com/JanaLasser/school_SEIRX/releases/tag/v1.0.0.

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

## Acknowledgements

J.S. and S.T. acknowledge financial support from the Austrian Science Promotion Agency FFG under 882184 and J.L. and P.K. from the Vienna Science and Technology Fund WWTF under MA16-045. PK and ST are grateful for support from the Medizinisch-Wissenschaftlichen Fonds des Bürgermeisters der Bundeshauptstadt Wien, no. CoVid004. We are greatly indebted to the Carinthian Department of Education and to numerous principals and teachers supporting this work. We thank Wolfgang Knecht for help with the visualization. The computational results presented have been achieved in part using the Vienna Scientific Cluster (VSC).

## Author contributions

J.L. Designed the study, wrote the code for the simulations, analysed the simulations, wrote the original draft of the manuscript and conducted interviews with school stakeholders. J.S. implemented the online visualisation tool, conducted interviews with school stakeholders and provided valuable feedback for the manuscript. L.R. collected, cleaned and provided the cluster data and provided valuable feedback for the manuscript. S.T.

Provided valuable feedback on the manuscript. D.S. Supervised the conduction of the study and provided valuabel feedback on the manuscript. P.K. Designed the study, supervised the conduction of the study, wrote the original draft of the manuscript and provided valuable feedback for the manuscript. All authors have read and edited the final version of the manuscript.

## Competing interests

The authors declare no competing interests.

## Additional information

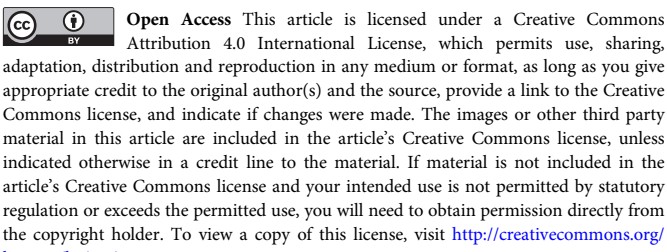

