## [Peer Review File · Nature Communications]

Reviewers' Comments:

Reviewer #1:

Remarks to the Author:

This paper investigates the impact of several different types of interventions on SARS-CoV-2 transmission in schools, using a relatively sophisticated agent-based model that is parameterized with empirical data from Austria. They find that combined interventions can be enough to keep cluster size down in schools, and that multiple interventions are more important for secondary schools than for primary schools. The research question is important to public health and the model and presentation are both of high quality. I like the model, but I have a few comments and concerns that the authors could address in revision:

1. I strongly encourage the authors to add vaccination to their model. The \$6 million dollar question for this fall is how/whether to re-open schools. There is now enough data on vaccine efficacy against infectivity to address this question. In particular, decision-makers are interested in how much vaccination is required to avoid socially and psychologically damaging interventions like hybrid online/in-person models where students attend every other week in person, or where they only attend half days every day. Conceivably, the combination of vaccination and less impactful interventions like testing and mask use could enable a safe return to full in-person learning. The alterations to the authors' model required to address this would be straightforward.
2. The role of age in transmission is a controversial topic, as the authors have pointed out. They have also raised some specific concerns such as selection bias that occurs when outbreak investigations start from symptomatic (older) individuals. But on several fronts, I am not convinced the authors have addressed this knowledge gap adequately in their analysis:
 - a. Their model assumes that transmission risk declines with age y as $0.02(18-y)$. However, this assumption was not varied in sensitivity analysis, and could directly drive their finding that outbreak control in primary schools is easier than control in secondary schools. They should add a sensitivity analysis where transmission risk does not vary with age.
 - b. In the same vein, I don't understand how the authors can state "secondary schools are a riskier transmission setting than primary schools" as a conclusion of their study, given that this finding was baked into their model assumptions via the equation $0.02(18-y)$. To me, this seems more like a premise, not a conclusion. Unless they can confirm this with sensitivity analysis such as I describe in point (a) above, this conclusion should be removed from the paper.
 - c. There has been much confusion and misinterpretation on the role of age, both in the public and even among many infectious disease modellers. Refs. 12-15 are oft-cited papers on the role of age in susceptibility. However, susceptibility, as these references define it, is the probability of infection per contact. Because older children have more contacts than adults (confirmed by empirical data such as the POLYMOD study, Mossong et al 2003, PLOS Bio), and younger children probably have more *effective* contacts than adults (e.g., little children have a hard time remembering to do physical distancing), it turns out that the infection rates in children are about as high as they are in adults. I had to write extensively on this question as part of the review process for previous papers--I hope no one minds if I cut and paste it here:
 - i. More recent studies in a broader range of populations estimate age-specific infection rates by measuring SARS-CoV-2 antibody seroprevalence. This has the advantage that under-reporting is no longer an issue. For instance, a study of household transmission in Barcelona, Spain found that child and adult household contacts were equally susceptible to infection (17.6% vs. 18.7%) [1]. A study of children from Bavaria, Germany showed that positive antibody tests in children were 6 times higher than PCR tests suggested, and that young and older children were equally likely to be infected [2]. A seroprevalence study in Brazil found that children were actually more likely to have been infected than individuals over 60 years of age [3]. Similarly, an antibody study in primary and high school children from several populations in Belgium found that when community transmission is high, young and older children are equally affected [4]
 - ii. Moreover, even antibody testing may under-estimate the infection rates in children because of differing antibody responses in children versus adults. Children generate more spike (S) antibodies than nucleocapsid (N) antibodies, but antibody tests often test only for N antibodies [5]. Similarly, the fact that children often fight the virus off more quickly than adults means less viral replication, and more false negatives in children than adults, when using antibody testing [5].
 - iii. Aside from evidence from antibody seroprevalence surveys, data from larger and more recent contact tracing studies than were available for the analyses in Davies et al and Sun et al support

that children get infected as often as adults. A study of 84,965 cases and 575,071 contacts in India found that contacts of child index cases and adult index cases were similarly likely to be infected [6]. This is echoed by a CDC study of household transmission suggesting children and adults are equally likely to transmit [7]. Finally, a recent analysis of the effectiveness of NPIs shows that opening/closing schools has a strong effect on the R value--comparable in magnitude or stronger than other interventions such as closing workplaces or limiting the sizes of gatherings--this would not be the case if susceptibility of children was significantly smaller than in adults [8]. More evidence along these lines is summarized in [9].

iv. Taken together, these more recent results suggest that children are infected at much higher rates than was thought earlier in the pandemic, and this was probably missed because estimates of cases in children based on PCR testing are significantly under-reported.

1. <https://academic.oup.com/cid/advance-article/doi/10.1093/cid/ciaa1721/5979490>.
2. <https://www.sciencedirect.com/science/article/pii/S2666634020300209>
3. [https://www.thelancet.com/journals/langlo/article/PIIS2214-109X\(20\)30387-9/fulltext](https://www.thelancet.com/journals/langlo/article/PIIS2214-109X(20)30387-9/fulltext)
4. https://www.sciensano.be/sites/default/files/limburg-validation-sars-cov2_report_20201112_final.pdf
5. <https://www.nature.com/articles/s41590-020-00826-9>
6. <https://science.sciencemag.org/content/370/6517/691>
7. https://www.cdc.gov/mmwr/volumes/69/wr/mm6944e1.htm?s_cid=mm6944e1_w
8. <https://www.sciencedirect.com/science/article/pii/S1473309920307854>
9. <https://onlinelibrary.wiley.com/doi/full/10.5694/mja2.50823>

Apropos the current paper, It was not clear whether this distinction between susceptibility and contact number/effectiveness was accommodated in their model. They do construct a contact network from class sizes, but unless I missed something, the contact network does not appear to be weighted for the possibility of effective contacts/frequency of contacts. Although the existing data may not permit this, it does suggest that they should more strongly qualify their findings about the role of age in infection control.

3. Do the authors assume that a classroom is closed if a child or teacher tests positive? I could not find that in the model description.

Reviewer #2:

Remarks to the Author:

The authors study the transmission dynamics of SARS-COV-2 in schools and evaluate the effect of non-pharmaceutical interventions. They start from reported cluster data from to inform an agent-based simulator for Austrian primary and secondary schools. This study on the safety and implications of school opening during the COVID-19 pandemic is timely and relevant. I do have some fundamental remarks on the manuscript.

Warranty is required when interpreting the cluster data for causality. Observing two cases in the same class that can be linked through the infectious period, does not imply that there has been transmission. In addition, if the infections are caused by a transmission event, there is no guarantee that this occurred at school. It is possible that these cases are linked via their infected parents, or not even linked at all. How is this addressed in the analysis/manuscript?

As mentioned in the discussion, it is possible or likely that schools mirror the infection dynamics in the general population. As such, transmission at school might not drive the epidemic and prevention measures can have limited impact. Using more data on outbreaks to inform the model does not solve this issue (line 66 and following). Especially given my previous point.

The analysis is based on 2-level mixing (household and school level) but lacks community mixing, if I understood correctly. All infections of adult agents (except from partners within one household) can only occur be though the infection of children at school. With this model design, the transmission at school is an important route, and has a high potential to block transmission. Please elaborate more on this and/or capture out-of-school mixing in the analysis.

The authors conclude that ventilation has most effect on the prevention of infections. This seems logical given their model assumptions in which they attribute most effect of ventilation. The

rationale on the ventilation parameter is clearly described in the Methods section. However, this is also theoretic and the manuscript would benefit from a more nuanced message.

In the sensitivity analysis, the authors focus on the implementation of the measures in practice. However, this sensitivity analysis also challenges the model assumption on the measures. This analysis shows that the conclusions are highly sensitive on the implementation, and as such on the model assumptions. I miss this nuance in the results and abstract. All results and recommendations are conditional on the assumptions with respect to the measures. To guide policy making, it is important to include and stress model-based and parametric uncertainty.

The sub-analysis for a variant with increased transmission potential is interesting though it seems not applicable for current VOCs given the model assumption of a fully susceptible population at the start of each simulation. More than one year after the start of the COVID19 pandemic, disease history is likely to play a role. This is not included in the model, which is fine, but also not discussed in the paper, which seems a shortcoming.

The school types are not fully clear to me. Why are the "secondary" school specifications in Table 2 not the combination of the lower and upper secondary schools? The number of classes (in Table 2) are easy to interpret, but why is there an increase in class size and especially in the number of teachers?

For the calibration (line 1000 and following), the authors specified ranges (start, stop, step) for 3 parameters, which end up in 950 combinations. From these combinations they randomly sampled 100 sets to continue their calibration. This approach does not guaranty to cover the full range of the parameter space. It is possible to sample 100 times (almost) the same values of for example c_1 . Why did the authors not adjust their step size so all combinations are feasible to run? As such, the full parameter range will be covered by design.

The authors created a nice visualization tool which is accessible online and their code is open-source. This is definitely an added value of the analysis.

Reviewer #3:

Remarks to the Author:

This manuscript deals with a very timely topic, namely, how prevention measures against the spread of sars-cov2 could make the opening of schools possible, while avoiding the formation of clusters. The manuscript first quickly analyzes data from clusters in Austrian schools. It leverages these data to calibrate an individual-based model for the spread of sars-cov2 in schools, and then considers the impact of several measures on this spread: face masks, ventilation, cohorting, testing. The simulations consider several types of schools, and the main results are that single measures are most often not enough: combinations of measures are typically necessary. Moreover, increased transmissibility or poor execution of measures could lead to strongly increased cluster sizes.

Overall, this manuscript deals with an interesting topic. However, the presentation of the assumptions and results is rather poor, making it difficult to apprehend and hiding the role of many assumptions, the corresponding limitations and impact on results. The model and the implementation of measures have so many parameters that one expects that many results will strongly depend on the situation, and giving very general results might be misleading. In particular, giving specific numbers in the abstract seems too strong a claim. Moreover, the data and model are specific to the Austrian case: the data used is from clusters in Austria, the structure of the schools is built specifically for Austrian schools, so all the results concern Austria, and there is no guarantee that they could be extended to other countries, where in particular class sizes, number of classes, structures of teaching schedules, number of teachers per class might be very different. This specificity should be made clear in the title/abstract.

I list below a number of concerns and points that would need improvement, in no particular order. Note that a lot of details and explanations that would be necessary to understand the main text

are left to the Methods part, and that some of these points directly stem from this issue.

-figure 1: what does "inconclusive" means?

-it should be written from the start and in a much clearer way that the school contact network are models and not data. The manuscript is ambiguous on this point, letting the reader believe that the agent-based model is based on contact data. As these models are based in part on a series of assumptions, this clearly limits the applicability of the study.

-the calibration method should also be explained more in the main text, and in particular what are the specific parameters. It seems there are (at least) 4 parameters (in addition to some assumptions about the way in which transmissibility decreases with age); it is quite worrying that no errorbars on the results are given, nor is there any study on how much the result depend on specific assumptions on the school contact networks structures.

-from the calibration and data point of view: shouldn't the model also reproduce the probability of not having a cluster in a school? This is quite important as the data contains only cases of clusters, by definition, but the model itself will have many instances where no cluster is present. Hopefully these instances are actually the majority even in reality so this is a crucial point

-it seems that the data is analyzed and used for calibration under the assumption that no measure was in place; is this confirmed, was there no mask wearing at schools? In other countries mask wearing has quickly become mandatory in schools so I find this strange.

-the calibration and all results clearly depend on the assumed structure of school network contacts. From this point of view, the assumption that the close contacts are only between students sharing a table seems limited, as students have friends with whom they spend more time. Also, the assumption that contact strengths do not depend on age seems a limitation, adolescents being probably less in very close contact than very young children. Overall, the impact of the modeling assumptions on the school network are not clear enough.

-I do not really understand the role of household members in the network. Are they there only to provide additional contagion paths?

-the authors write that the model is calibrated to reproduce "(iv) the ratio of teacher-to-student source cases"; however from the calibrating method, it seems that this is actually an input.

-the baseline scenario seems to include a reactive screening of everyone in the school. Is this really what was implemented in reality? It seems a very broad testing that would be rather costly.

-the effect of masks (table 1) seems quite conservative, see <https://www.pnas.org/content/118/4/e2014564118> where references finding a stronger effect are discussed

-the results section is very tedious to read, with lists of numbers for cases and subcases. Most importantly, it is very tedious to find the parameters corresponding to each measure. These should be clearly given when discussing the results.

-results are given for primary, lower secondary, secondary schools, but the differences between these schools should be made clearer. Is it just the size? Results for upper secondary are barely given. Probably it would be clearer to give only for primary and secondary, leaving to supplementary material lower/upper secondary and maybe writing only whether there are important differences with secondary.

-in figures 3-4, it is very interesting to show the whole distributions. However, the observed R or average cluster size should also be written next to each curve. This would make it much easier for the reader to compare the effect of the various measures.

-in K.3, there are some numbers whose meaning is very unclear: where does the "speaking

volume" enter into the model, for instance?

-for the reduced class size: a fraction of students is home every second day. I understand how this work for 50%, but how does it work for lower fractions?

-lines 248-252 are unclear: "except for ..." but then the list is of half of the possible measures, so the term "Except" is not appropriate

-Moreover, the results obviously depend on the parameter values for each measure. In particular, the ranking of the effectiveness of results clearly depend on these as well. The fact that one or 2 or 3 measures are needed also depend on these choices. This should be discussed much more, as these parameters are in part arbitrary, with large uncertainties on their actual values.

-risk transmission increases with age, the contrary is written lines 436-438 (right column): "the risk for transmission increases by about 25% upon con- tact with a six year old person compared to a contact with an 18 year old person"

-the equation before line 700 lacks parenthesis around $(1-q_i)$

General remarks

We thank the reviewers for their constructive feedback. We are glad to see that they see merit in our approach and for receiving valuable suggestions for how to further improve the quality of our manuscript. We have taken each comment thoroughly into account and, following the reviewer's suggestions, introduced three major changes to the manuscript.

(1) Calibration: We revised our approach to the calibration. After calibrating the transmission risk for household contacts separately, we first did a coarse search of the full parameter space of the intermediate contact weight c_1 , the loose contact weight c_2 and the reduction of transmission risk due to younger age, c_3 with a low number of simulations per parameter combination (N=500). Importantly, we also removed the constraint $c_1 > c_2$. We found that in the most optimal calibration solutions $c_2 > c_1$, which is unrealistic, as this would signify that the transmission risk in less intense contact situations (duration, proximity) is higher than in more intense ones. This result indicated that our initial assumptions about contact types that arise in the school setting did not fit well with reality. To address this discrepancy, we chose to change the contact intensity of all contacts that arise between students and teachers (teaching and daycare supervision) from being intermediate to loose. After introducing this change, we found a stable optimal solution for c_1 , but c_2 did not converge to a stable value anymore. This makes sense since after introducing this change, only a small number of contacts of intermediate intensity remained, which apparently weren't enough to influence the system enough to significantly drive infection dynamics. Apparently, the empirical data about outbreaks in schools accessible to us and the lack of real-world measurements of contact intensities leads to a situation in which we do not have enough information to confidently calibrate a system that differentiates within-school contacts. We therefore decided to not differentiate between different contact types in the school context anymore and only retain a single contact weight $c_1 = c_2$ for school contacts. We note that we renamed the calibrated parameters into $c_{contact}$ and c_{age} for clarity in the manuscript and the following description.

Following the coarse-grained parameter search, we performed a fine-grained parameter search to determine the values of $c_{contact}$ and c_{age} with a high number of simulations per parameter combination (N=4000).

To assess the uncertainty associated with the optimal parameter combination we implemented a bootstrapping approach on the simulated ensembles: We randomly (with replacement) sampled 2000 out of the 4000 simulation runs for every parameter

combination, calculated the difference E between the empirically measured and simulated (subsampled) outbreak size distributions and determined the parameter combination for $c_{contact}$ and c_{age} that minimized E for a total of 1000 samplings. We use the median values for $c_{contact}$ and c_{age} as calibrated values for all subsequent simulations and perform a sensitivity analysis, using the 0.025 and 0.975 percentile values for each of the two parameters. The sensitivity analysis shows that the model is robust to changes in $c_{contact}$ and c_{age} within this range (see Fig. S3).

To ensure that the thus determined optimal parameter combination was not subject to large changes if different distance measures were used to calculate E , we introduced additional distance measures between the empirical and simulated distributions of outbreak sizes: in addition to the sum of squared differences we calculated the χ^2 distance, the Bhattacharyya distance, the Pearson and Spearman correlation (we use the absolute value of the difference between 1 and the correlation coefficient as distance measure), and the slopes of pp- and qq-plots (we use the absolute value of the difference between 1 and the slope as distance measure). We show the heatmaps of values for the different distance measures and different parameter combinations in Fig. S1 in the SI. The median values and [0.025; 0.975] confidence intervals are shown in Fig. S2. While there is some variation between the different distance measures, all confidence intervals overlap and we confirm that the optimal parameter choice only slightly depends on the distance measure.

This updated calibration procedure is described in the Materials and Methods, section “Calibration details”.

(2) Delta variant: We now use the epidemiological parameters of the B.1.617.2 or “delta” strain of the virus, since this strain has become dominant across the globe at the time of writing. Specifically, we assume a transmission risk that is increased by a factor of 2.25 as compared to the original strain of the virus (Liu and Rocklöv 2021 and Campbell et al. 2021). In addition, following recent reports (Zhang et al. 2021), we assume a reduced incubation period duration of 4.4 ± 1.9 days (original strain: 6.4 ± 1.9 days). To match the reduced incubation period, we also adjust the duration of the latent period to 3.0 ± 1.9 days (original strain: 5.0 ± 1.9 days). Since, to our knowledge, there are no reports about a change in the duration of infectiousness in the literature, we leave it at 10.91 ± 3.95 days. The parameters of this strain are used in all simulations except the calibration simulations, as those are matched to outbreaks with the original strain.

(3) Vaccinations: We introduce the possibility of agents to be vaccinated, which gives them limited immunity against infection with the virus. Specifically, we model the effectiveness of

the BNT162b2 and ChAdOx1 vaccines 90 days after the second dose of vaccination, i.e. including a drop in effectiveness with time. Pouwels et al. 2021 report a 78% and 61% effectiveness against high viral load of BNT162b2 and ChAdOx1, respectively. As our population will have a mix of different vaccines and durations since the second dose, we chose a conservative estimate of vaccine effectiveness of 60% in our simulations. Since the viral load of vaccinated people infected with the delta variant seems to be similarly high as the load of unvaccinated people (Brown et al., 2021), we do not assume a lower infectiousness of vaccinated people, if they get infected. We introduce a vaccination scenario, resembling the current reality in Austria, in which 80% of teachers are vaccinated (ORF.at, September 2021), 60% of the general population (i.e. family members in our model) are vaccinated (Austrian Ministry for health, September 12, 2021) 0% of children < 15 (i.e. students in our model) are vaccinated (Infopoint Coronavirus, September 12, 2021). We also model a second scenario in the near future, in which vaccines for children are approved and significant numbers of children (50%) are vaccinated. Information regarding vaccinations is also described in the Materials and Methods, section “Vaccinations” (lines 865 to 893).

Below we provide a detailed point-by-point response to reviewer comments. We strongly believe that with these changes we can now address all issues in a satisfactory way.

Reviewer #1

(1) I strongly encourage the authors to add vaccination to their model. The \$6 million dollar question for this fall is how/whether to re-open schools. There is now enough data on vaccine efficacy against infectivity to address this question. In particular, decision-makers are interested in how much vaccination is required to avoid socially and psychologically damaging interventions like hybrid online/in-person models where students attend every other week in person, or where they only attend half days every day. Conceivably, the combination of vaccination and less impactful interventions like testing and mask use could enable a safe return to full in-person learning. The alterations to the authors' model required to address this would be straightforward.

Following the reviewer's suggestion, we added two vaccination scenarios (i) 80% of teachers, 60% of family members and 0% of students vaccinated, and (ii) 80% of teachers, 60% of family members and 50% of students are vaccinated (see also point (3) in our general remarks above). In addition we now assume the delta variant (B.1.617.2) to be the dominant strain, as is currently (September 2021) the case in Europe (see also point (2) in our general remarks above). For vaccination efficacy, we assume a 60% efficacy against high viral load. To our knowledge, this reflects a conservative estimate of vaccine efficacy, based on the most recent reports of vaccination efficacy for the ChAdOx1 and BNT162b2 vaccines against delta, 90 days after the second dose (Liu and Rocklöv 2021 and Campbell et al. 2021). In addition, we assume no reduction in transmissibility once a vaccinated person has become infected, reflecting preliminary reports of viral load in vaccinated and unvaccinated people being equal (CDC, 2021). Results of simulations with partially vaccinated populations in addition to conservative estimates for measure effectiveness are shown in Fig. 5. The impact of vaccinations on infection dynamics in addition to the measure combinations with the original (optimistic) parameter choices is shown in figure S6.

(2) The role of age in transmission is a controversial topic, as the authors have pointed out. They have also raised some specific concerns such as selection bias that occurs when outbreak investigations start from symptomatic (older) individuals. But on several fronts, I am not convinced the authors have addressed this knowledge gap adequately in their analysis:

a. Their model assumes that transmission risk declines with age y as $0.02(18-y)$. However, this assumption was not varied in sensitivity analysis, and could directly

drive their finding that outbreak control in primary schools is easier than control in secondary schools. They should add a sensitivity analysis where transmission risk does not vary with age.

Following the reviewer's concern, we added a sensitivity analysis for the age dependence of the transmission risk. We compare outbreak sizes in the calibrated baseline model ($c_{age} = -0.005$, optimistic assumptions for measure effectiveness parameters) with the 0.025 and 0.975 percentile values of c_{age} from the calibration. Results are shown in Figure S4 and described in the beginning of the results section (line 194). We find that our model is robust to these changes in c_{age} : If the age dependence of the transmission risk is $[-0.0225; 0.0]$, average outbreak sizes (averaged over all school types and investigated intervention scenarios) change by a factor of $[0.92, 1.03]$. The change in outbreak sizes in primary schools is slightly more pronounced ($[0.89, 1.04]$) than the change in secondary schools ($[0.96, 1.02]$). Nevertheless, these changes are small when compared to changes in assumed measure effectiveness and do not change the observation that outbreaks in secondary schools are much larger than outbreaks in primary schools.

b. In the same vein, I don't understand how the authors can state "secondary schools are a riskier transmission setting than primary schools" as a conclusion of their study, given that this finding was baked into their model assumptions via the equation $0.02(18-y)$. To me, this seems more like a premise, not a conclusion. Unless they can confirm this with sensitivity analysis such as I describe in point (a) above, this conclusion should be removed from the paper.

As already mentioned above, the finding that transmission risk declines with age is not baked into our model but a result of our calibration procedure and thereby informed by data. In principle, it would also be possible that the coefficient (-0.005) would turn out to be zero. Furthermore, as we tried to stress in the manuscript, the main difference between secondary and primary schools is not necessarily this age-dependent transmission risk but the structural difference in the school-specific contact networks. As described in our reply to point (2) above, the sensitivity analysis of the age dependent transmission risk shows that it is only responsible for a small difference in outbreak sizes between primary and secondary schools and does not drive the main result we describe.

One limitation that could arise from our calibration procedure (and that we already discuss in the limitations) is that the calibrated model reflects biases in the cluster tracing data due to a higher likelihood of asymptomatic infections in younger children. By explicitly including the fact that children are less likely to develop symptoms upon infection in our model and

thereby, e.g., trigger TTI we aim to remove this bias from the calibration as far as technically possible.

*c. There has been much confusion and misinterpretation on the role of age, both in the public and even among many infectious disease modellers. Refs. 12-15 are oft-cited papers on the role of age in susceptibility. However, susceptibility, as these references define it, is the probability of infection per contact. Because older children have more contacts than adults (confirmed by empirical data such as the POLYMOD study, Mossong et al 2003, PLOS Bio), and younger children probably have more *effective* contacts than adults (e.g., little children have a hard time remembering to do physical distancing), it turns out that the infection rates in children are about as high as they are in adults. I had to write extensively on this question as part of the review process for previous papers--I hope no one minds if I cut and paste it here:*

i. More recent studies in a broader range of populations estimate age-specific infection rates by measuring SARS-CoV-2 antibody seroprevalence. This has the advantage that under-reporting is no longer an issue. For instance, a study of household transmission in Barcelona, Spain found that child and adult household contacts were equally susceptible to infection (17.6% vs. 18.7%) [1]. A study of children from Bavaria, Germany showed that positive antibody tests in children were 6 times higher than PCR tests suggested, and that young and older children were equally likely to be infected [2]. A seroprevalence study in Brazil found that children were actually more likely to have been infected than individuals over 60 years of age [3]. Similarly, an antibody study in primary and high school children from several populations in Belgium found that when community transmission is high, young and older children are equally affected [4]

ii. Moreover, even antibody testing may under-estimate the infection rates in children because of differing antibody responses in children versus adults. Children generate more spike (S) antibodies than nucleocapsid (N) antibodies, but antibody tests often test only for N antibodies [5]. Similarly, the fact that children often fight the virus off more quickly than adults means less viral replication, and more false negatives in children than adults, when using antibody testing [5].

iii. Aside from evidence from antibody seroprevalence surveys, data from larger and more recent contact tracing studies than were available for the analyses in Davies et al and Sun et al support that children get infected as often as adults. A study of 84,965 cases and 575,071 contacts in India found that contacts of child index cases

and adult index cases were similarly likely to be infected [6]. This is echoed by a CDC study of household transmission suggesting children and adults are equally likely to transmit [7]. Finally, a recent analysis of the effectiveness of NPIs shows that opening/closing schools has a strong effect on the R value--comparable in magnitude or stronger than other interventions such as closing workplaces or limiting the sizes of gatherings--this would not be the case if susceptibility of children was significantly smaller than in adults [8]. More evidence along these lines is summarized in [9].

iv. Taken together, these more recent results suggest that children are infected at much higher rates than was thought earlier in the pandemic, and this was probably missed because estimates of cases in children based on PCR testing are significantly under-reported.

1. <https://academic.oup.com/cid/advance-article/doi/10.1093/cid/ciaa1721/5979490>.

2. <https://www.sciencedirect.com/science/article/pii/S2666634020300209>

3.

[https://www.thelancet.com/journals/langlo/article/PIIS2214-109X\(20\)30387-9/fulltext](https://www.thelancet.com/journals/langlo/article/PIIS2214-109X(20)30387-9/fulltext)

4.

https://www.sciensano.be/sites/default/files/limburg-validation-sars-cov2_report_2020_1112_final.pdf

5. <https://www.nature.com/articles/s41590-020-00826-9>

6. <https://science.sciencemag.org/content/370/6517/691>

7. https://www.cdc.gov/mmwr/volumes/69/wr/mm6944e1.htm?s_cid=mm6944e1_w

8. <https://www.sciencedirect.com/science/article/pii/S1473309920307854>

9. <https://onlinelibrary.wiley.com/doi/full/10.5694/mja2.50823>

Apropos the current paper, It was not clear whether this distinction between susceptibility and contact number/effectiveness was accommodated in their model. They do construct a contact network from class sizes, but unless I missed something, the contact network does not appear to be weighted for the possibility of effective contacts/frequency of contacts. Although the existing data may not permit this, it does suggest that they should more strongly qualify their findings about the role of age in infection control.

We thank the reviewer for this comprehensive and illuminating summary of this branch of literature. In this modelling work we take an agnostic approach to this issue by making no a priori assumptions on the strength of a potential age effect as outlined above. As described in Methods (subsection Transmission Risk), different types of contact (household / school) come with different transmission risks as determined per model calibration. Furthermore, contact networks are school-type specific. Therefore, both the number of contacts and quality of these contacts vary as a function of age. The age-dependence of the transmission risk therefore reflects the effective impact of age on transmission risk that stems from all mechanisms that are not explicit in the model (which contains age-group-specific contact networks and probabilities for symptomatic infections).

Thank you also for pointing out that adherence to and effectiveness of certain measures like keeping distance might vary with age. We have added a discussion of this point in the limitations of the study, stating that the age-dependent transmission risk might also be influenced by such factors. Note that if younger children show less adherence to certain measures one would naively expect to observe an increase of the transmission risk with decreasing age as a result of our calibration procedure.

(3) Do the authors assume that a classroom is closed if a child or teacher tests positive? I could not find that in the model description.

In the calibration scenario, if a child or teacher tests positive, the classroom is closed. This reflects the policy in autumn 2020 when teachers and students did not wear masks during lessons in Austria. In the measure simulation scenarios, classrooms are not closed if a teacher or student tests positive. This reflects the policy enacted in Austria starting in February 2021. We initially described this policy in the Materials and Methods (lines 967 to 971) and expanded on this explanation to clarify the policy in the simulation.

Reviewer #2

(1) Warrantly is required when interpreting the cluster data for causality. Observing two cases in the same class that can be linked through the infectious period, does not imply that there has been transmission. In addition, if the infections are caused by a transmission event, there is no guarantee that this occurred at school. It is possible that these cases are linked via their infected parents, or not even linked at all. How is this addressed in the analysis/manuscript?

The definition of in-school transmission is as follows.

Definition of a source (index) case of a school cluster: a case with the earliest date of onset among a school cluster, presumed the source case is generated outside the school setting (e.g. household, friends, transport) or at least the source case is not generated in the school setting by another case.

Definition of successive cases: occurrence of the case in the time period between the minimum and maximum of the serial interval (estimated based on the Austrian surveillance data) and a known infectious contact to the source case (defined as above) and exclusion of any other probable source case outside the school.

This very conservative definition should reduce the likelihood of misclassification of school transmission.

(2) As mentioned in the discussion, it is possible or likely that schools mirror the infection dynamics in the general population. As such, transmission at school might not drive the epidemic and prevention measures can have limited impact. Using more data on outbreaks to inform the model does not solve this issue (line 66 and following). Especially given my previous point.

We do fully agree. The model based recommendations for preventive measures should reduce the risk of transmission in the school setting for the school population. Based on the assumption that this age group does not drive the epidemic, the school-related preventive measures will have a limited impact only on community transmission. Nevertheless, it should avoid school closures in the upcoming autumn and help decision makers choose the right measure combinations for schools.

(3) The analysis is based on 2-level mixing (household and school level) but lacks community mixing, if I understood correctly. All infections of adult agents (except from

partners within one household) can only occur be though the infection of children at school. With this model design, the transmission at school is an important route, and has a high potential to block transmission. Please elaborate more on this and/or capture out-of-school mixing in the analysis.

We focus indeed on modeling transmissions in schools. Concretely, we address the research question of how likely an index case in a school setting (be it student or teacher, irrespective of how this primary infection was acquired) will lead to further transmissions in the school. We do not address the extent to which these cases might lead to infections in other settings than schools and thereby contribute to the infection dynamics observed in the entire population. We have added a statement in the introduction clarifying that out-of-school mixing is beyond the scope of our analysis (line 157).

(4) The authors conclude that ventilation has most effect on the prevention of infections. This seems logical given their model assumptions in which they attribute most effect of ventilation. The rationale on the ventilation parameter is clearly described in the Methods section. However, this is also theoretic and the manuscript would benefit from a more nuanced message.

Several of our assumptions concerning the effectiveness of interventions like ventilation are based on values reported in the literature. We interpret several of these values of “optimistic” in the sense that this is what could be achieved in terms of effectiveness if the measure were properly implemented and strictly adhered to. To communicate these uncertainties properly, we include a sensitivity analysis making more conservative assumptions regarding the effectiveness of individual measures and reporting that the same set of measures might lead to up to a 157-fold increase in outbreak size if the measures were poorly executed. We have revised the manuscript to ensure that these limitations are stressed in all sections of the manuscript and whenever we draw policy-relevant conclusions from our modelling results.

(5) In the sensitivity analysis, the authors focus on the implementation of the measures in practice. However, this sensitivity analysis also challenges the model assumption on the measures. This analysis shows that the conclusions are highly sensitive on the implementation, and as such on the model assumptions. I miss this nuance in the results and abstract. All results and recommendations are conditional on the assumptions with respect to the measures. To guide policy making, it is important to include and stress model-based and parametric uncertainty.

We agree with the referee's observation regarding the sensitivity of our results; see also our response to the previous remark. We have revised the abstract and result section to ensure a more nuanced reporting of the results and recommendations.

(6) The sub-analysis for a variant with increased transmission potential is interesting though it seems not applicable for current VOCs given the model assumption of a fully susceptible population at the start of each simulation. More than one year after the start of the COVID19 pandemic, disease history is likely to play a role. This is not included in the model, which is fine, but also not discussed in the paper, which seems a shortcoming.

We now include two vaccination scenarios in our simulations (see also point (3) in our general remarks). Since a previous infection will result in a similar immune response, we think that by including these vaccination scenarios we also cover scenarios in which significant parts of the population have a disease history.

(7) The school types are not fully clear to me. Why are the “secondary” school specifications in Table 2 not the combination of the lower and upper secondary schools? The number of classes (in Table 2) are easy to interpret, but why is there an increase in class size and especially in the number of teachers?

The school types chosen for the study are the most prevalent school types in Austria. The number of classes, students per class and teachers per school were extracted from the most recent national school statistics. Lower secondary schools, upper secondary schools and secondary schools have different organizational structures and different teaching schedules in Austria, which is also reflected in their different number of classes, class sizes and number of teachers. Importantly, secondary schools are *not* just a summary statistic for lower secondary schools and upper secondary schools. They are a different school type with a different syllabus and different students, which we reflect in our school type models.

(8) For the calibration (line 1000 and following), the authors specified ranges (start, stop, step) for 3 parameters, which end up in 950 combinations. From these combinations they randomly sampled 100 sets to continue their calibration. This approach does not guaranty to cover the full range of the parameter space. It is possible to sample 100 times (almost) the same values of for example c_1 . Why did the authors not adjust their step size so all combinations are feasible to run? As such, the full parameter range will be covered by design.

Following the reviewer's suggestion we adapted the calibration process. We now perform a coarse grid search over the full parameter range, followed by a more fine grained grid search around the optimum of the coarse grid search (see also a detailed description of the revised calibration process in point (1), general remarks and the updated description in the Materials & Methods, lines 1061 and following).

Reviewer #3

(1) Overall, this manuscript deals with an interesting topic. However, the presentation of the assumptions and results is rather poor, making it difficult to apprehend and hiding the role of many assumptions, the corresponding limitations and impact on results. The model and the implementation of measures have so many parameters that one expects that many results will strongly depend on the situation, and giving very general results might be misleading. In particular, giving specific numbers in the abstract seems too strong a claim. Moreover, the data and model are specific to the Austrian case: the data used is from clusters in Austria, the structure of the schools is built specifically for Austrian schools, so all the results concern Austria, and there is no guarantee that they could be extended to other countries, where in particular class sizes, number of classes, structures of teaching schedules, number of teachers per class might be very different. This specificity should be made clear in the title/abstract.

We have changed the title to reflect that our results primarily concern Austrian schools, and also state the Austria-context once more in the abstract. We thank the referee for pointing out that it was not clear in the abstract whether the results stem from the cluster tracing data or the model. We have changed the abstract to make it clear whether we are describing the data or results of the model. Actually, almost all of the specific numbers we give in the abstract were obtained by analyzing the cluster tracing data, so we believe that it is appropriate to also give the key facts in the abstract. Specific numbers from results of the model (other than more qualitative descriptions of what are the main findings from the modelling) are given only once in the abstract, namely when we want to highlight that our modeling results are sensitive with regard to the assumptions we made in the model, which appears to be in line with the point that the reviewer makes in this comment. Finally, regarding sensitivity, it is clear that the outbreak size (measured in cumulative cases) is particularly sensitive to parameters and also shows substantial variations for the same parameter setting, as we describe in the manuscript. It is also clear that our findings regarding how different prevention measures influence the effective reproduction number are

more robust for the simple reason that additive errors in the effective reproduction number roughly result in multiplicative errors in the outbreak size. Note that our main results concern which combination of prevention measures is needed to reduce R_{eff} below one, as is stated in the beginning of the abstract.

Furthermore, we have added a statement in which we briefly compare how Austrian class sizes and student-to-teacher ratios compare to other countries (lines 645 to 651). In brief, Austrian class sizes and student-to-teacher ratios are close to the EU average according to statistics from the OECD which suggests that at least in that regard school-type-specific contact networks in Austria are comparable in size to the networks in other European countries.

(2) I list below a number of concerns and points that would need improvement, in no particular order. Note that a lot of details and explanations that would be necessary to understand the main text are left to the Methods part, and that some of these points directly stem from this issue.

We tried to improve the clarity of the main text by adding additional information that was previously mentioned only in the Methods part throughout.

(3) figure 1: what does "inconclusive" means?

"inconclusive" is based on the definition given in Section I of the Materials and Methods (Empirical observations of SARS-CoV-2 clusters in Austrian schools):

Data on the exact school type was not available at the time of our analysis, therefore we assigned the clusters based on the age of students using the following algorithm:

1. primary: the age of all affected students is ≤ 10 years.
2. lower secondary: the age of all affected students is within the interval $[10,15]$ years.
3. upper secondary: the age of all affected students is ≥ 15 years.
4. secondary: the age of all affected students is ≥ 10 years.
5. inconclusive: otherwise.

To clarify this, we now include a reference to the appropriate Materials and Methods section in the legend of Fig. 1.

-it should be written from the start and in a much clearer way that the school contact network are models and not data. The manuscript is ambiguous on this point, letting the reader believe that the agent-based model is based on contact data. As these models are based in part on a series of assumptions, this clearly limits the applicability of the study.

We have now also added an explicit statement in the introduction (line 109) that the properties of the contact networks are modelled from educational statistics when we first mention the use of these networks.

-the calibration method should also be explained more in the main text, and in particular what are the specific parameters. It seems there are (at least) 4 parameters (in addition to some assumptions about the way in which transmissibility decreases with age); it is quite worrying that no errorbars on the results are given, nor is there any study on how much the result depend on specific assumptions on the school contact networks structures.

We now included a brief summary of our calibration procedure in the introduction (line 121). We now also provide error bars (mean \pm SD) from bootstrapping simulation results with every value when we describe the results of the calibration (lines 184 to 217, see also our general remarks, point (1) for details on changes in the calibration procedure).

Regarding the impact of different contact network structures on the dynamics of infection spread, the different school types provide insight into the impact of structural changes, as each school type has a quite different contact network structure regarding the network's density and number of nodes. In addition, we now also include the possibility for additional contacts through friendships in our sensitivity analysis (line 364) and discuss them in the manuscript alongside the other dimensions of our sensitivity analysis. Such additional contacts are now also included in our "conservative" scenario (Figure 5).

-from the calibration and data point of view: shouldn't the model also reproduce the probability of not having a cluster in a school? This is quite important as the data contains only cases of clusters, by definition, but the model itself will have many instances where no cluster is present. Hopefully these instances are actually the majority even in reality so this is a crucial point

As our model is initiated with the introduction of a case into the school, there is always a cluster in the model and the question is whether this cluster will be detected or not. Indeed, it might be the instance that these source cases are not detected and/or lead to no other

transmissions in schools that are detected and this might happen in the model as well as in reality. As the cluster tracing data does not contain such events, we cannot validate the number of these instances. It is nevertheless true that in our calibrated model, a significant number of simulations do not lead to any follow-up cases (63% for primary schools and primary schools with daycare, 44% for lower secondary schools, 55% for lower secondary schools with daycare, 44% for upper secondary schools and 40% for secondary schools). We now note these numbers in the Materials and Methods section (lines 1231 to 1238).

-it seems that the data is analyzed and used for calibration under the assumption that no measure was in place; is this confirmed, was there no mask wearing at schools? In other countries mask wearing has quickly become mandatory in schools so I find this strange.

Mask-wearing was mandatory when, e.g., entering or leaving the classroom but not during the lessons while students sit at their desks. The measure we evaluate in the manuscript therefore refers to wearing masks also during the lessons. We clarify that now in the introduction and repeat it several times throughout the manuscript that we evaluate a mask policy during lessons assuming that masks are compulsory when students are not sitting at their desks.

-the calibration and all results clearly depend on the assumed structure of school network contacts. From this point of view, the assumption that the close contacts are only between students sharing a table seems limited, as students have friends with whom they spend more time.

After revising our calibration method, we do not differentiate between contacts of varying strength within the school anymore. We came to the conclusion that our empirical data is not sufficient to reliably differentiate between contacts of different intensity within a school. We now only retain “household” contacts and “school contacts”, which have a transmission risk that is reduced by 70% as compared to household contacts (following our calibration, see also our general remarks, point (1)). Students in a given class have a fully connected contact network, which therefore also covers all potential friendships between students in the given class. We nevertheless recognize the limitation regarding additional contacts between friends from different classes. We therefore incorporated an approximation of additional contacts between friends of different classes in our model and investigated its impact in the sensitivity analysis (Figure S2). We also incorporated this in our analysis of a model with very conservative assumptions: in our conservative model, 20% randomly chosen students

of every class have an additional contact to a random student from a random class that is different from the student's class.

Also, the assumption that contact strengths do not depend on age seems a limitation, adolescents being probably less in very close contact than very young children. Overall, the impact of the modeling assumptions on the school network are not clear enough.

We agree that contact strengths might depend on age and thank the reviewer for pointing out this important relationship. We incorporate this into our model by specifically allowing for a changing transmission risk with age – a factor that is calibrated against empirical observations of outbreaks. The age dependence of the transmission risk is a compound factor, which represents a summary of all factors that might influence transmission risk and vary with age, for example biological factors such as differences in viral load, differences in height and therefore distribution of aerosols, lung volume, and also contact strength. The empirical data that is accessible to us lacks the detail to untangle these factors and calibrate them separately. Since we find only a 0.5% reduction of transmission risk per year, we conclude that the possible increase in contact strength for younger children is outweighed by other factors. We nevertheless now mention an age dependence of contact strength as a possible contributor to the overall age dependence of transmission risk (line 824). In addition, we adapted the text in the Materials and Methods to more clearly describe the modelling assumptions that shape the contact networks (line 979 and following).

-I do not really understand the role of household members in the network. Are they there only to provide additional contagion paths?

Yes, household members provide additional contagion paths by which infections can spread from one class to the next in case two siblings of the same household go to the same school, as it is frequently the case in Austria where efforts are undertaken to enable siblings to attend the same school.

-the authors write that the model is calibrated to reproduce "(iv) the ratio of teacher-to-student source cases"; however from the calibrating method, it seems that this is actually an input.

This is true – we corrected the misleading formulation.

-the baseline scenario seems to include a reactive screening of everyone in the school. Is this really what was implemented in reality? It seems a very broad testing that would be rather costly.

The reactive screening of everyone is definitely something that was implemented during the calibration period (autumn 2020). Austria was one of the very few countries in the world which stringently tested K2 contacts of positive cases and invested significant amounts of money and effort into implementing a high-performance testing infrastructure. Teachers and students in the same class as a positive case were considered K1 contacts (table neighbours) or K2 contacts (rest of the classmates and the teacher) and therefore also tested with PCR tests. After school re-opening in February, this policy changed since the definition of K1 and K2 contacts changed, and there is no default reactive screening of the whole class after a positive test anymore. We therefore incorporate reactive screening only in the calibration simulations, but not in all other simulations. We clarified this in the Materials and Methods section (lines 904 to 915).

-the effect of masks (table 1) seems quite conservative, see <https://www.pnas.org/content/118/4/e2014564118> where references finding a stronger effect are discussed

We note that we differentiate between the transmission risk reduction if the transmitting agent wears a mask (50%), and the transmission risk reduction if the receiving agent wears a mask (30%). Both reductions have a multiplicative effect on overall transmission risk reduction. Therefore, if both the transmitting and the receiving agent wear a mask, transmission risk is reduced by 85%. Given the PNAS review, this actually seems to be an optimistic estimate. We now also give the combined number and reference to the PNAS study (line 862) to contextualize the parameter values chosen for our study.

-the results section is very tedious to read, with lists of numbers for cases and subcases. Most importantly, it is very tedious to find the parameters corresponding to each measure. These should be clearly given when discussing the results.

We have rewritten the results section to make it less list-like. Next to referencing the sources for the parameter estimates once more, we also reference the assumptions and/or used parameter values explicitly in the result section when first mentioning a measure.

-results are given for primary, lower secondary, secondary schools, but the differences between these schools should be made clearer. Is it just the size? Results for upper secondary are barely given. Probably it would be clearer to give

only for primary and secondary, leaving to supplementary material lower/upper secondary and maybe writing only whether there are important differences with secondary.

We have restructured the result section accordingly, focussing on primary and secondary schools only. On most occasions we now give numbers for the outbreak sizes or R , respectively, instead of giving both values. We have also added a section that summarizes how school types differ. In brief, they differ in the structure of their contact networks (size & density) and in the age-dependent transmission risks. We compare these differences now also quantitatively to make it easier to grasp how much which of these effects actually contributes in terms of transmission risk.

-in figures 3-4, it is very interesting to show the whole distributions. However, the observed R or average cluster size should also be written next to each curve. This would make it much easier for the reader to compare the effect of the various measures.

We now show mean numbers of R next to each distribution.

-in K.3, there are some numbers whose meaning is very unclear: where does the "speaking volume" enter into the model, for instance?

We report the parameter choices which were used as inputs into the COVID transmission risk calculator¹ to reflect the situation in a classroom. The risk calculator outputs a single scalar number for the reduction of transmission risk, given a certain scenario. The transmission risk reduction linearly depends on the input parameters. A speaking volume of 3 is characterized as "loud speaking" by the authors of the calculator, which seemed to be a fitting choice of a teacher teaching a class.

-for the reduced class size: a fraction of students is home every second day. I understand how this work for 50%, but how does it work for lower fractions?

For lower fractions, a random number of students equal to the specified fraction is removed from the class on any given school day.

-lines 248-252 are unclear: "except for ..." but then the list is of half of the possible measures, so the term "Except" is not appropriate

Indeed. We have reformulated this paragraph.

¹ <https://www.mpic.de/4851094/risk-calculator> and <https://www.mdpi.com/1660-4601/17/21/8114/htm>

-Moreover, the results obviously depend on the parameter values for each measure. In particular, the ranking of the effectiveness of results clearly depend on these as well. The fact that one or 2 or 3 measures are needed also depend on these choices. This should be discussed much more, as these parameters are in part arbitrary, with large uncertainties on their actual values.

We don't agree that our parameters are arbitrary; they are either based on literature or on our calibration procedure; see Table 1. We agree that they partially come with large uncertainties which we proactively address from the start of the manuscript in the abstract (see replies above). We already discussed this issue under limitations but added it once more at the beginning of the discussion section.

-risk transmission increases with age, the contrary is written lines 436-438 (right column): "the risk for transmission increases by about 25% upon contact with a six year old person compared to a contact with an 18 year old person"

Thank you for spotting this, we corrected it.

-the equation before line 700 lacks parenthesis around $(1-q_i)$

Thank you for spotting this, we corrected it.

Reviewers' Comments:

Reviewer #1:

Remarks to the Author:

The authors have responded to my comments in a satisfactory way. I have no other changes to suggest.

Reviewer #2:

Remarks to the Author:

The authors did a great job in revising their manuscript according all comments and feedback from both reviewers. Their point-by-point reply with a general update on the calibration, delta variant and vaccination is detailed and comprehensive. The other replies are constructive and respectful.

It is not clear to me how the authors handled the discrepancy that the reference data (=observed clusters in 2020) is based on infections with the "wild type" SARS-COV-2, while their simulation study for late 2021 targets transmission dynamics based on the Delta strain. Is this captured in the model calibration and application?

A small remark, the last sentence of the abstract is not clear to me "However, large clusters might still occur on an infrequent, however, regular basis."

Reviewer #3:

Remarks to the Author:

The authors have addressed all my comments in a satisfactory way and the manuscript has improved a lot. I recommend publication.

Reviewer # 2

(1) It is not clear to me how the authors handled the discrepancy that the reference data (=observed clusters in 2020) is based on infections with the “wild type” SARS-COV-2, while their simulation study for late 2021 targets transmission dynamics based on the Delta strain. Is this captured in the model calibration and application?

We clarified our approach in the respective part of the introduction (second-to-last paragraph), which now reads:

“To adapt the calibrated model to a situation in which the delta variant is prevalent, we multiply the household transmission risk – which was calibrated using data from outbreaks of the original strain – by 2.25, reflecting the drastically increased infectivity of the variant (19, 20) and adapt the incubation period to 4.4 ± 1.9 days (21). We also adapt the latent period to 3.0 ± 1.9 days.”

We also refer the reviewer to the Methods section (“Calibration and parameter choices” -> “Household contacts”), where we describe the calibration procedure and the adjustments made to reflect the epidemiological parameters of the delta variant in more detail.

(2) A small remark, the last sentence of the abstract is not clear to me “However, large clusters might still occur on an infrequent, however, regular basis.”

We have shortened the abstract considerably (was 329 words, is now 150 words). We have removed the sentence the reviewer refers to in the process. The new abstract reads as follows:

“We aim to identify those measures that effectively control the spread of SARS-CoV-2 in Austrian schools. Using cluster tracing data we calibrate an agent-based epidemiological model and consider situations where the B.1.617.2 (delta) virus strain is dominant and parts of the population are vaccinated to quantify the impact of non-pharmaceutical interventions (NPIs) such as room ventilation, reduction of class size, wearing of masks during lessons, vaccinations, and school entry testing by SARS-CoV2-antigen tests. In the data we find that 40% of all clusters involved no more than two cases, and 3% of the clusters only had more than 20 cases. The model shows that combinations of NPIs together with vaccinations are necessary to allow for a controlled opening of schools under sustained community transmission of the SARS-CoV-2 delta variant. For plausible vaccination rates, primary (secondary) schools require a combination of at least two (three) of the above NPIs.”